# SHIFTING TIME: TIME-SERIES FORECASTING WITH KHATRI-RAO NEURAL OPERATORS

## ABSTRACT

We present an operator-theoretic framework for time-series forecasting that involves learning a continuous time-shift operator associated with temporal and spatio-temporal problems. The time-shift operator learning paradigm offers a continuous relaxation of the discrete lag factor used in traditional autoregressive models enabling the history of a function up to a given time to be mapped to its future values. To parametrize the operator learning problem, we propose Khatri-Rao neural operators – a new architecture for defining non-stationary integral transforms which achieves almost linear cost on spatial and spatio-temporal problems. From a practical perspective, the advancements made in this work allow us to handle irregularly sampled observations and forecast at super-resolution in both space and time. Detailed numerical studies across a wide range of temporal and spatio-temporal benchmark problems suggest that the proposed approach is highly scalable and compares favourably with state-of-the-art methods.

## 1 INTRODUCTION

Time series forecasting is a fundamental problem in machine learning and statistics with applications to a broad spectrum of problems encountered in all branches of science, engineering, and finance (Roberts et al., 2013; Milani et al., 2017; Siami-Namini and Namin, 2018). At a high-level, time-series problems are concerned with forecasting the future values of quantities of interest given past observations of the same or correlated quantities.

The majority of methods for time-series forecasting largely fall into the categories of autoregressive moving average models and their variants (Box and Jenkins, 1976; Girard, 2004), and deep autoregressive models with memory (El-

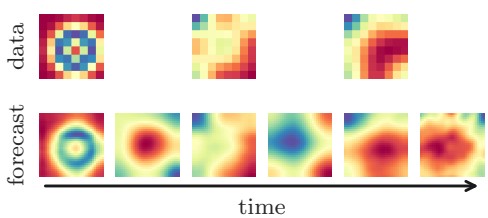

Figure 1: The top row shows low-resolution test data. In the bottom row we plot a high-resolution forecast. By parametrizing the time-shift operator by a Khatri-Rao neural operator we can forecast in super-resolution in both *space* and *time*.

man, 1990; Hochreiter and Schmidhuber, 1997; Salinas et al., 2020). With the tremendous success of transformer-based models in natural language processing tasks (Vaswani et al., 2017) and computer vision applications (Dosovitskiy et al., 2020), this class of models are gaining popularity in time-series forecasting (Chen et al., 2021; Zhou et al., 2022; Wu et al., 2022; Liu et al., 2022; 2024; Gruver et al., 2024). In the world of spatio-temporal forecasting, Gaussian processes (Hamelijnck et al., 2021), deep operator networks (DeepONets) (Lu et al., 2021), and neural operators (Li et al., 2020a;b;c) have emerged as cornerstones of the literature.

A major challenge with all autoregressive-style models is that observations are required to be provided at a constant frequency at both training and inference time. This requirement introduces a number of challenges in practice. First, when observations are not provided at regular intervals, it is common practice to create a hierarchy of approximations which can negatively impact performance for reasons unrelated to the capacity of the model. Second, in an online setting, this requirement will necessitate the creation of a pipeline for imputing any missing datapoints (due to sensor error or system latency) before predictions can be made. While neural ordinary differential equations have shown tremendous

promise for learning from irregularly spaced observations (Chen et al., 2018; Rubanova et al., 2019), they are challenging to scale and train for large scale temporal and spatio-temporal datasets.

In the present work, we propose casting time-series forecasting problems as a supervised learning problem of the *continuous time-shift operator*. In contrast to standard autoregressive models based on discrete-time (or discrete space-time) representation of the dynamics, the continuous time-shift operator maps the entire, continuous history of the dynamics over a past time-window into its future values over a subsequent time-window. Our operator-theoretic approach can be viewed as a continuous relaxation of the discrete lag factor in autoregressive models. This provides several practical advantages such as the ability to learn directly from irregularly sampled observations and to forecast at super-resolution in both space and time while retaining the stability of training neural operators; see Figure 1.

In order to deal with the complexities of learning the time-shift operator for temporal and spatio-temporal dynamical systems, we propose Khatri-Rao neural operators (KRNOs). KRNOs are a new architecture for operator learning based on non-stationary integral transforms which provides exceptional model flexibility compared to methods based on stationary kernels (Li et al., 2020c), while achieving almost linear scaling. In Section 3 we demonstrate the efficacy of the proposed approach on a suite of challenging test cases including the Darts datasets (Herzen et al., 2022), the M4 datasets (Makridakis et al., 2020), shallow water simulation (Kissas et al., 2022), and a climate modeling problem (Kissas et al., 2022). In total, we consider 27 different test cases and compare performance against 22 modern approaches for temporal and spatio-temporal forecasting to demonstrate the strong generalization capabilities of the proposed approach.

## 2 METHOD

We first introduce the continuous time-shift operator for temporal and spatio-temporal dynamical systems. Following this, we propose Khatri-Rao neural operators for learning the time-shift operator.

### 2.1 THE CONTINUOUS TIME-SHIFT OPERATOR

Consider an ordinary differential equation (ODE) $\dot{z}(t) = F(z(t)), z(0) = z_0$, with Lipschitz continuous $F : \mathbb{R}^n \to \mathbb{R}^n$ over the time-interval $[0, T]$. In contrast to the flow map that maps the initial condition to the solution at time $t$, here we consider a causal, continuous-time operator that translates the history of $z$ over $[t_p, t]$ into its future values over $(t, t_f]$, where $0 \leq t_p < t < t_f \leq T$. We refer to this operator as the *time-shift operator* that can be written as a propagator of the form

$$z(\tau) = (\mathcal{A}_{t_p}^{t,t_f} z)(\tau), \ \forall \tau \in (t, t_f]. \tag{1}$$

The existence of $\mathcal{A}_{t_p}^{t,t_f} : L^2([t_p, t]; \mathbb{R}^n) \to L^2((t, t_f]; \mathbb{R}^n)$ follows from the Picard-Lindelöf theorem and noting that $z(\tau) = z(t) + \int_t^\tau F(z(s))ds, \ \tau \in (t, t_f)$. We would like to highlight two key properties of the time-shift operator: (1) semigroup property: $\mathcal{A}_{t_p}^{t_2,t_f} = \mathcal{A}_{t_1}^{t_2,t_f} \circ \mathcal{A}_{t_p}^{t_1,t_2}$, where $t_p < t_1 < t_2 < t_f$, and (2) continuity property: $\exists C > 0$ such that $||A_{t_p}^{t,t_f} z_1 - A_{t_p}^{t,t_f} z_2||_{L^2((t,t_f];\mathbb{R}^n)} \leq C||z_1 - z_2||_{L^2([t_p,t];\mathbb{R}^n)}$ for all $z_1, z_2 \in L^2([t_p, t]; \mathbb{R}^n)$; see Appendix A for a proof.

Since the time-shift operator defined in (1) is a continuous-time operator, it can be learned from datasets with irregularly sampled observations (similar to neural ODEs (Chen et al., 2018) but without requiring adjoint ODE based sensitivity calculations), which is a significant advantage in many practical applications. Moreover, since the operator depends on $t_p$ and $t_f$, this representation enables the dynamics of complex systems to be studied over different time scales. In the context of time-series forecasting, we treat $t_p$ and $t_f$ as hyperparameters that can be learned from data using cross-validation or inferred using hyper-gradients. Finally, another practical advantage associated with the time-shift operator is that it enables super-resolution forecasts since it is a continuous-time model.

The notion of shift-operators has been widely studied in functional analysis (Marchenko, 2006). Recent theoretical work (Zhen et al., 2021; 2022) leveraged time-shift operators while studying the relationship between the spectra of the autocorrelation function and the infinite-dimensional Koopman operator (Koopman, 1931a) governing the evolution of observables. However, to the best of our knowledge, the idea of developing a operator-theoretic framework to directly learn the continuous time-shift operator from time-series data has not been explored before.

In the present work, we propose to parametrize the time-shift operator using a neural operator. To motivate this, consider the special case when $\mathcal{A}_{t_p}^{t,t_f} : L^2([t_p, t]; \mathbb{R}^n) \to L^2((t, t_f]; \mathbb{R}^n)$ is a Hilbert-Schmidt operator (Retherford, 1993). Then there exists a kernel $\kappa : [0, T] \times [0, T] \to \mathbb{R}$ satisfying the condition $\int_0^T \int_0^T |\kappa(\tau, s)|^2 ds d\tau < \infty$ such that

$$z(\tau) = (\mathcal{A}_{t_p}^{t,t_f} z)(\tau) = \int_{t_p}^t \kappa(\tau, s) z(s) ds, \quad \forall \tau \in (t, t_f], \tag{2}$$

where the dependence of the kernel on $(t, t_p, t_f)$ is not explicitly indicated for simplicity of notation. It is worth noting that even though the preceding continuous convolution integral representation holds under restrictive assumptions on the dynamics, it motivates the application of deep neural operators involving a nested composition of integral transforms and point-wise operations to approximate the time-shift operator of general nonlinear dynamical systems.

We can similarly define the spatio-temporal time-shift operator for a scalar field $u : \Omega \times [0, T] \to \mathbb{R}$, where $\Omega \subset \mathbb{R}^{d-1}$, $d > 1$ denotes a bounded Lipschitz domain. Using the non-overlapping time-intervals defined previously, the spatio-temporal time-shift operator can be defined as

$$u(x, \tau) = (\mathcal{A}_{t_p}^{t,t_f} u)(x, \tau), \quad \forall x \in \Omega, \ \tau \in (t, t_f]. \tag{3}$$

Under the assumption that $u$ lies in the separable Hilbert space $\mathcal{U}(\Omega \times [0, T]; \mathbb{R})$ and the spatio-temporal time-shift operator is a Hilbert-Schmidt operator that maps from $\mathcal{U}(\Omega \times [t_p, t]; \mathbb{R})$ to $\mathcal{U}(\Omega \times (t, t_f]; \mathbb{R})$, we have the following integral representation

$$u(x, \tau) = (\mathcal{A}_{t_p}^{t,t_f} u)(x, \tau) = \int_\Omega \int_{t_p}^t \kappa(\{x, \tau\}, \{y, s\}) u(y, s) dy ds, \ x \in \Omega, \ \tau \in (t, t_f], \tag{4}$$

where $\kappa : \Omega \times [0, T] \times \Omega \times [0, T] \to \mathbb{R}$ is a square integrable kernel. The preceding representation in terms of an integral transform motivates the application of deep neural operators to approximate the time-shift operator of complex spatio-temporal dynamical systems. Furthermore, universal approximation results for neural operators (Kovachki et al., 2023) ensure that under appropriate regularity assumptions, the time-shift operator can be well approximated.

It is worth noting that Li et al. (2020c) considered an operator learning test problem where a two-dimensional flow-field over the time-interval $[0, 10]$ is mapped to $(10, T]$ (with fixed $T$). They tackled this using an autoregressive Fourier neural operator (FNO) model and a 3D FNO model which makes predictions over the entire spatio-temporal domain of interest. The time-shift operator learning formalism presented here allows us to view the test-case involving FNO-3D in (Li et al., 2020c) as a special case of the general setting considered here with a stationary-kernel based neural operator parametrization of the time-shift operator and fixed values of $(t_p, t_f)$. In the next section, we present a new architecture for parametrizing the time-shift operator that enables over an order of magnitude reduction in the number of parameters compared to FNO, while achieving superior accuracy.

## 2.2 Khatri-Rao neural operators (KRNOs)

We now introduce KRNOs, a new operator learning architecture based on non-stationary integral transforms, to approximate the time-shift operator of temporal and spatio-temporal dynamical systems. KRNOs offer expressive parametrization of operators using non-stationary integral transform layers which (i) do not require any approximation of the kernel and (ii) scale almost linearly in the number of quadrature nodes. As far as we are aware, ours is the only approach for parametrizing neural operators which combines these advantages. We will show later that KRNOs provide state-of-the-art performance across a number of benchmarks while inheriting the benefits of neural operators such as being discretization independent and enabling super-resolution in forecasts (Li et al., 2020c).

**Neural operators** Neural operators (NOs) are an expressive class of models for approximating maps between function spaces. In contrast to standard multi-layer perceptrons, which are defined by an alternating series of affine maps and nonlinear activations, NOs are defined by an alternating

---

[1]By "Exact Kernel" we mean that the only source of error in our methodology comes from the quadrature scheme, with the non-stationary kernel evaluated without additional approximation errors.

Table 1: Comparison of Graph Neural Operator (Li et al., 2020a), Multipole Graph Neural Operator (Li et al., 2020b), and Fourier Neural Operator (Li et al., 2020c), for computing kernel integral transforms, as compiled by (Kovachki et al., 2023). Here $N' << N$ is a constant used to control the variance of the integral transform approximation. Ours is the only approach which allows for exact, non-stationary kernel evaluations while achieving almost linear computational cost.

| Method | Time | Non-stationary | Exact kernel[1] |
|--------|------|----------------|--------------|
| Graph Neural Operator | $\mathcal{O}(NN')$ | ✓ | ✗ |
| Multipole Graph Neural Operator | $\mathcal{O}(N)$ | ✓ | ✗ |
| Fourier Neural Operator | $\mathcal{O}(N \log N)$ | ✗ | ✓ |
| Khatri-Rao Neural Operator (ours) | $\mathcal{O}(N^{1+1/d})$ | ✓ | ✓ |

series of linear, kernel integral transforms and nonlinear activations. For simplicity of exposition, consider an integral transform layer (Li et al., 2020c; Kovachki et al., 2023) that maps the input spatio-temporal vector field $v_\ell : \Omega \times [0, \tau] \to \mathbb{R}^p$ to $v_{\ell+1} : \Omega \times [0, \tau] \to \mathbb{R}^q$, defined below

$$v_{\ell+1}(t, x) = \mathcal{K}(v_\ell)(t, x) = \int_\Omega \int_0^\tau \kappa(\{t, x\}, \{t', x'\}) v_\ell(t', x') dt' dx' + W v_\ell(t, x) + b, \quad (5)$$

where $\kappa : \mathbb{R} \times \Omega \times \mathbb{R} \times \Omega \to \mathbb{R}^{q \times p}$ is a matrix-valued kernel, $W \in \mathbb{R}^{q \times p}$ is a weight matrix, and $b \in \mathbb{R}^q$ is a bias vector. It is also common to prepend and append the preceding layer by a series of point-wise lifting and projection layers (Kovachki et al., 2023). Note that in (5), the inputs and outputs are assumed to be defined over the same spatio-temporal domain for simplicity – we will later consider the general case when the input and output domains are different.

Rather than computing the integral transforms exactly, NOs propagate evaluations of the intermediate functions at a set of quadrature nodes through the network. Let $X = \{\{t_1, x_1\}, \{t_2, x_2\}, \ldots, \{t_N, x_N\}\} \in \mathbb{R}^{N \times d}$, where $t_i \in \mathbb{R}$ and $x_i \in \mathbb{R}^{d-1}$, denote the set of $N$ quadrature nodes in the full spatio-temporal domain ($N = n^d$, where $n$ is the number of quadrature nodes per dimension) and let $w \in \mathbb{R}^N$ denote the vector of quadrature weights. As we will show, while computing the point-wise transformation defined by the weights $W$ and $b$ scales as $\mathcal{O}(qpN)$, the primary computational bottleneck in computing the output from a kernel integral transform layer arises from approximating the integral over the domain of the input function.

For the discussion on computational complexity that follows we omit writing the dependence on $q$ and $p$ since these are architectural considerations and the required value for $N$ will be dependent on the complexity of the input function. Letting $v_\ell(X) \in \mathbb{R}^{N \times p}$ be the $\ell^{\text{th}}$ layer evaluated at the quadrature nodes, the kernel integral transform can be approximated as

$$\int_\Omega \int_0^\tau \kappa(X, \{t', x'\}) v_\ell(t', x') dt' dx' \approx \kappa(X, X) \text{vec}(\text{diag}(w) v_\ell(X)), \quad (6)$$

where $\text{vec} : \mathbb{R}^{N \times p} \to \mathbb{R}^{Np}$ creates a vector from a matrix by stacking columns, $\text{diag} : \mathbb{R}^N \to \mathbb{R}^{N \times N}$ converts a vector into a diagonal matrix, $\kappa(X, \{t', x'\}) \in \mathbb{R}^{Nq \times p}$ represents the kernel evaluated between all the quadrature nodes $X$ in the output domain and a single node $\{t', x'\}$ in the input domain. Meanwhile, $\kappa(X, X) \in \mathbb{R}^{Nq \times Np}$ is the kernel evaluated between all the quadrature nodes $X$ in both the output and input domains. Clearly this approach scales as $\mathcal{O}(N^2)$ which is prohibitively expensive for even a modest number of quadrature nodes. In light of these computational challenges a number of approaches have been developed including Graph Neural Operators (Li et al., 2020a), Multipole Graph Neural Operators (Li et al., 2020b), and Fourier Neural Operators (FNOs) (Li et al., 2020c). As we will show, our approach is the only one which scales almost linearly in the number of quadrature nodes while enabling non-stationary integral transforms with exact kernel evaluations.

**Khatri-Rao product structure** In order to achieve almost linear scaling in $N$ without having to approximate the kernel function, we assume that the kernel function decomposes as a product,

$$\kappa(\{t, x\}, \{t', x'\}) = \kappa^{(1)}(t, t') \odot \left( \odot_{i=2}^d \kappa^{(i)}([x]_{i-1}, [x']_{i-1}) \right), \quad (7)$$

where $\kappa^{(i)} : \mathbb{R} \times \mathbb{R} \to \mathbb{R}^{q \times p}$ for $i = 1, \ldots, d$, $\odot$ denotes the element-wise product, and $[x]_i \in \mathbb{R}$ indicates the $i^{\text{th}}$ element of $x$. While this assumption may appear limiting at first glance, it has been

applied extensively in the context of Gaussian process (GP) regression to build new positive definite kernels and to scale GP regression on product grids (Saatçi, 2011; Wilson et al., 2014). For example, the squared exponential kernel, the Matérn class of kernels, and the spectral mixture product kernel all decompose as a product of the form in Equation (7).

**Proposition 1.** *If the quadrature nodes lie on a product grid, $X = \bar{t} \times x^{(1)} \times \ldots x^{(d-1)}$, where $\bar{t} \in \mathbb{R}^n$ and $x^{(i)} \in \mathbb{R}^n$ denote the quadrature nodes along the time dimension and the $i^{th}$ dimension of the spatial coordinate $x$, respectively, and the kernel function has a component-wise product structure of the form given in Equation (7), then the kernel function evaluated at the quadrature nodes inherits the Khatri-Rao product structure,*

$$\kappa(X, X) = \kappa^{(1)}(\bar{t}, \bar{t}) * \left( \underset{i=2}{\overset{d}{*}} \kappa^{(i)}(x^{(i-1)}, x^{(i-1)}) \right), \tag{8}$$

*where $\kappa^{(i)}(\cdot, \cdot) \in \mathbb{R}^{qn \times pn}$ is a block-partitioned matrix where block $jk$ is the $jk^{th}$ output from the component kernel $\kappa^{(i)}$ evaluated on the outer product of the quadrature nodes along the $i^{th}$ dimension.*

Proposition 1 follows from similar results for Kronecker structured GP regression (Saatçi, 2011) (see Appendix B for details). A practical consequence of Proposition 1 is that computing matrix-vector products between between a Khatri-Rao structured matrix of size $qN \times pN$ and a vector of size $pN$ only requires $\mathcal{O}(N^{2/d} + N)$ storage and $\mathcal{O}(N^{1+1/d})$ time, without the need to explicitly form the full matrix of size $qN \times pN$ (see Appendix D). It is important to note that the parameters $p$ and $q$ are architectural parameters which are common to all NO approaches. Table 1 provides a comparison of KRNOs to other NOs in the literature. To reiterate what was mentioned previously, ours is the only approach which achieves almost linear cost while enabling non-stationary integral transforms without having to approximate the kernel function.

We would like to emphasize that Proposition 1 is valid even when the input and output domains are different with different resolution quadrature grids (see Appendix C for details). This approach can be viewed as a continuous analog of upsampling and downsampling techniques commonly used in convolutional neural networks. For applications with high spatial or temporal resolutions, we recommend using lower-resolution quadrature grids within the internal kernel integral layers as a practical means to significantly reduce computational cost and memory requirements. Furthermore, comprehensive details on the training and inference costs of KRNO and FNO across different spatial resolutions are presented in Tables 8,9, as well as Figure 9 in Appendix-G.

In this work, we parametrize each component-wise kernel, $\kappa^{(i)} : \mathbb{R} \times \mathbb{R} \to \mathbb{R}^{q \times p}$ by a neural network. More details on our parametrization can be found in Appendix E. To illustrate the efficacy of KRNO, we first applied our method to the Darcy-flow and hyper-elastic benchmark problems from Lu et al. (2022) and Li et al. (2023), respectively. Figure 2 illustrates the predictions from KRNO for these two problems; see Figure 11 in the appendix for additional details. It can be noted from Table 2 that KRNO provides improved performance over FNO (Li et al., 2020c) and DeepONet (Lu et al., 2021) on both problems. We will later benchmark the performance of KRNO on learning the time-shift operator across a variety of challenging datasets and benchmarks to demonstrate that our approach often provides competitive performance to or even outperforms SOTA methods.

## 2.3 PRACTICAL ASPECTS OF LEARNING THE TIME-SHIFT OPERATOR

Consider an $n-$dimensional multivariate discrete time-series dataset $\{z_t\}_{t=0}^T$. This dataset is first converted into pairs of input and output sequences over two non-overlapping time intervals $[t_p, t]$ and $(t, t_f]$, where $0 \le t_p < t < t_f \le T$, for various time instances $t$. The input sequence, denoted by $U_p^t = \{z_t\}_{t=t_p}^t$, includes $z_t$ given at $P$ time steps within the look-back window $[t_p, t]$. The output sequence, denoted by $U_f^t = \{z_t\}_{t=t}^{t_f}$, contain values of $z_t$ given at $H$ time steps within the prediction window $(t, t_f]$. These pairs of sequences, for different values of $t$, are used to approximate the continuous time-shift operator $A_{t_p}^{t,t_f}$ using KRNO. For estimating the KRNO parameters, we minimize the loss function, $\frac{1}{M} \sum_{i=1}^M ||U_f^{t_i} - \mathcal{A}_{t_p}^{t_i,t_f} U_p^{t_i}||_{L^2((t_i,t_f];\mathbb{R}^n)}$, where $M$ is the number of input-output sequence pairs. As discussed previously, $t_p$ and $t_f$ are hyperparameters of the time-shift operator which are chosen using cross-validation.

Similar to temporal datasets, we consider spatio-temporal data comprising discrete snapshots of spatial fields $\{u_t(x)\}_{t=0}^T$, where $x \in \Omega_g$ with $\Omega_g$ representing a spatial grid over $\Omega$. This dataset

Table 2: Performance comparison of different NOs on Darcy-flow and hyper-elastic problems. Results with $(\cdot)^{\dagger}$, $(\cdot)^{\ddagger}$ are from Lu et al. (2022) and Li et al. (2023), respectively.

| Method | $L^2$ relative error | |
| --- | --- | --- |
| | Darcy-flow | Elasticity |
| FNO | $1.19 \pm 0.05\%^{\dagger}$ | $5.08\%^{\ddagger}$ |
| DeepONet | $1.36 \pm 0.12\%^{\dagger}$ | $9.65\%^{\ddagger}$ |
| KRNO (ours) | $\mathbf{0.96 \pm 0.04\%}$ | $\mathbf{4.56\%}$ |

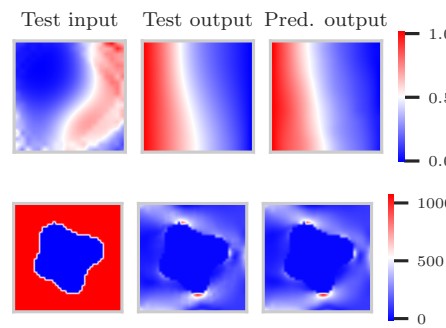

Test input    Test output   Pred. output

Figure 2: The top row presents a sample prediction from the test set for the Darcy-flow problem, while the bottom row illustrates a sample prediction for the elasticity problem.

is converted into pairs of input and output sequences of spatial fields, $U_p^t(x)$ and $U_f^t(x)$, over time intervals $[t_p, t]$ and $(t, t_f]$, where $0 \le t_p < t < t_f \le T$. The input sequence $U_p^t(x) = \{u_t(x)\}_{t=t_p}^t$ contains spatial fields corresponding to $P$ time steps within the look-back window $[t_p, t]$. The output sequence $U_f^t(x) = \{u_t(x)\}_{t=t}^{t_f}$ contains spatial fields over $H$ time steps within the prediction window $(t, t_f]$. These sequence pairs are used to learn the spatio-temporal time-shift operator (3) using KRNO by minimizing the loss function, $\frac{1}{M} \sum_{i=1}^M \|U_f^{t_i} - \mathcal{A}_{t_p}^{t_i, t_f} U_p^{t_i}\|_{L^2(\Omega) \times L^2((t_i, t_f])}$, where $M$ is the number of input-output sequence pairs.

## 3 NUMERICAL STUDIES

In this section, we evaluate the performance of the proposed time-shift operator on a suite of temporal and spatio-temporal forecasting problems. These datasets included two spatio-temporal datasets, and 16 diverse time-series datasets (8 univariate time series from Darts datasets, six datasets corresponding to different seasonalities from the M4 competition, one multivariate time series corresponding to trading prices of 14 cryptocurrencies, and a bi-variate time series containing the positions of NBA basketball players). In total, this amounts to 27 test cases. Across these test cases, we compare against 22 modern approaches for temporal and spatio-temporal forecasting problems.

For all the problems, we first convert the datasets into input and output sequence pairs as mentioned in the previous section. The default KRNO network architecture used in all the numerical studies has 128 channels in both the lifting and projection layers and three kernel integral transform layers. The component-wise kernel function used in KRNO is parameterized by a neural network with three hidden layers (see Appendix E). All the model hyperparameters are estimated through cross-validation (see Appendix H). Aggregated performance statistics of the proposed approach on all test cases are presented in Appendix Table 10. In Appendix F, we provide numerical studies demonstrating the performance of KRNO on time-series datasets with irregularly spaced observations.

Table 3: Comparison of the average relative $L^2$ errors on the shallow water problem for the three field variables.

| Method | $L^2$ relative error | | |
| --- | --- | --- | --- |
| | $\rho$ | $u$ | $v$ |
| FNO-3D | 0.00211 | 0.02606 | 0.02637 |
| LOCA | 0.00314 | 0.15221 | 0.14999 |
| KRNO | **0.00145** | **0.01497** | **0.01459** |

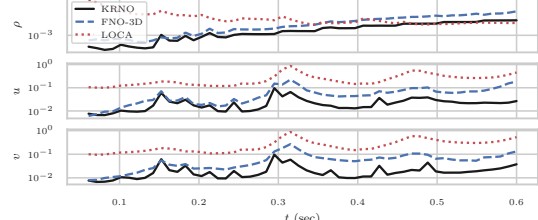

Figure 3: Comparison of the average relative $L^2$ errors as a function of time for the three field variables (across the 1000 test simulations) obtained using KRNO, FNO-3D and LOCA models.

## 3.1 SPATIO-TEMPORAL FORECASTING PROBLEMS

For spatio-temporal problems, we consider shallow water simulation (Kissas et al., 2022), and a climate modeling dataset (Kissas et al., 2022). For evaluation, we use the $L^2$ relative error metric, $L^2$ relative error $= ||u(\cdot,t) - \hat{u}(\cdot,t)||_{L^2(\Omega)}/||u(\cdot,t)||_{L^2(\Omega)}$, where $u(\cdot,t)$ and $\hat{u}(\cdot,t)$ are true and predicted spatial fields at time $t$.

**Shallow water example:** Here, the objective is to learn a spatio-temporal operator that is capable of predicting three field variables (fluid column height $\rho$, velocity in the $x_1$-direction $u$, and velocity in the $x_2$-direction $v$) over a future prediction window $(t, t_f]$ using historical data from a look-back window $[t_p, t]$, i.e., $\mathcal{U}(\Omega \times [t_p, t], \mathbb{R}^3) \to \mathcal{U}(\Omega \times (t, t_f], \mathbb{R}^3))$, where $\Omega := (0,1) \times (0,1)$ denotes the spatial domain. It is worth noting that the spatio-temporal forecasting problem statement considered here is significantly more challenging than the usual test case considered in previous studies (Kissas et al., 2022) which involves mapping the initial condition

Table 4: Comparison of trainable parameters.

| Method | #Parameters |
|---|---|
| FNO-2D | 466,075 |
| FNO-3D | 2,462,895 |
| LOCA | 94,477,220 |
| KRNO (ours) | 146,159 |

to the solution at a fixed time. The dataset used for this problem is taken from Kissas et al. (2022), which includes simulated data generated on a $32 \times 32$ spatial grid over the time window $(0,1)$ and collected at every 0.01 seconds. The training and testing datasets each consist of 1000 simulations with different initial conditions. Both the look-back and prediction window period are set to 0.05 seconds. For evaluation on testing data, we use the three field variables from the first 0.05 seconds window to recursively predict their evolution until 0.6 seconds. As a baseline method, we consider FNO-3D model (Kovachki et al., 2023) and attention based neural operator LOCA (Kissas et al., 2022) to approximate the time-shift operator alongside the proposed KRNO method. Table-3 and Figure 3 compares the relative $L^2$ error (averaged across 1000 test simulations) for the three field variables when training is conducted for 100 epochs. The results indicate that KRNO delivers superior performance relative to FNO-3D and LOCA. In addition, it is worth noting that KRNO only uses 6% of parameters required by FNO-3D (see Table 4). Predictions from KRNO for a test simulation are shown in Figure 4. Additional numerical results for this test case can be found in Appendix H.0.5.

**Climate modeling example:** In this experiment, we consider the problem of approximating a spatio-temporal time-shift operator that maps the surface air temperature and surface air pressure, i.e., $\mathcal{U}(\Omega \times [t_p, t]; \mathbb{R}^2) \to \mathcal{U}(\Omega \times (t, t_f]; \mathbb{R}^2))$, where $\Omega := [-90, 90] \times [0, 360]$ denotes the spatial domain defined in terms of latitude and longitude. The dataset is taken from Kissas et al. (2022) which is based on the Physical Sciences Laboratory meteorological data (Kalnay et al., 1996); see https://psl.noaa.gov/data/gridded/data.ncep.reanalysis.surface.html. The training data consists of daily temperature and pressure from 2000 to 2005 (1825 days) over a $72 \times 72$ spatial grid. The test contains observations from the years 2005 to 2010 on the same grid. The KRNO operator is trained on temperature and pressure data from a 7-day look-back window, with a matching 7-day prediction window. For the evaluation on testing data, we used data from the last week of the previous year and recursively predicted the temperature and pressure fields for the whole year. This is repeated for each year in the testing set. Representative predictions for pressure and temperature are shown in Figure 5 along with the corresponding relative $L^2$ errors. It can be seen that the proposed time-shift operator learning approach performs remarkably well for this dataset.

## 3.2 TEMPORAL FORECASTING PROBLEMS

We consider univariate and multivariate time-series data to evaluate the performance of the time-shift operator on temporal problems. The univariate datasets considered here are Darts (Herzen et al., 2022) and M4 (Makridakis et al., 2020). In the case of multivariate datasets, we benchmark our method using Crypto (Ticchi et al., 2021) and Player Trajectory datasets.[2]

One of the challenges in temporal forecasting using deep learning models is the presence of distribution shift in the data (Kouw and Loog, 2018; Wang et al., 2021; Kuznetsov and Mohri, 2020). A common practice to tackle distribution shifts is to use preprocessing strategies, which involve removing known trends and seasonality from the data. To handle distribution shifts in some datasets,

---

[2]https://github.com/linouk23/NBA-Player-Movements

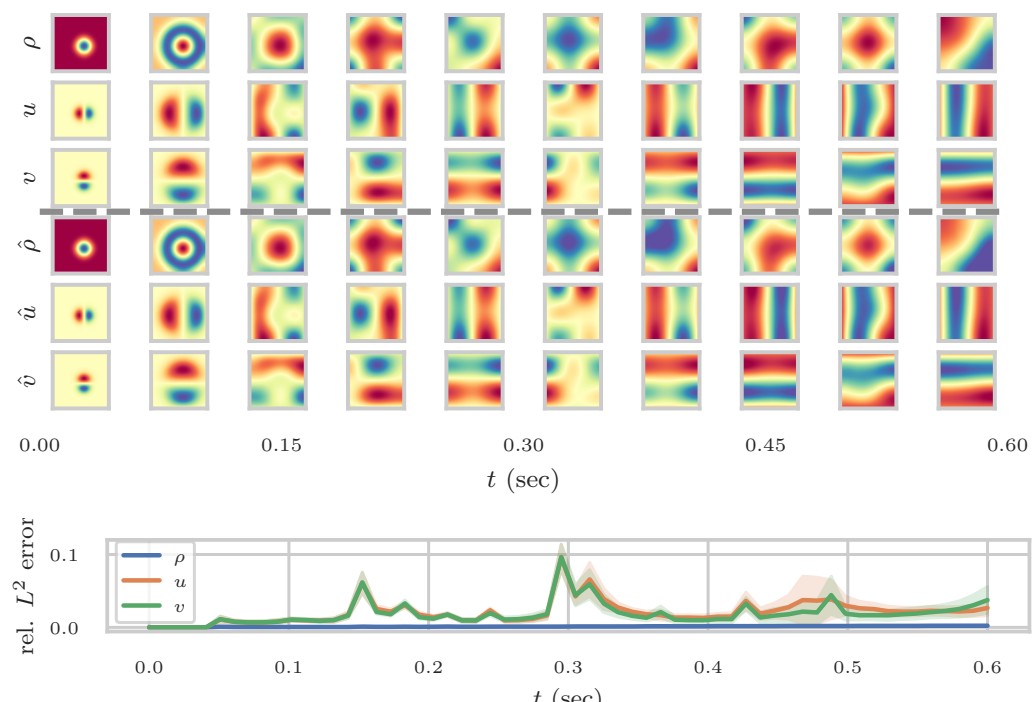

Figure 4: Shallow water problem: Top figure shows the predictions $(\hat{\rho}, \hat{u}, \hat{v})$ for the three field variables along with the true fields $(\rho, u, v)$ as a function of time for a test simulation using KRNO trained for 100 epochs. The bottom figure shows the error bars representing the $L^2$ relative errors for three field variables across the 1000 test simulations, with the shaded region indicating $\pm 1$ standard deviation. Additional error plots for the three fields are shown in Figure 13 in the Appendix.

we use reversible instance normalization ReVIN(Kim et al., 2021) to normalize each input sequence and denormalize the output sequences from the KRNO model.

**Darts benchmarks** We consider 8 univariate time-series datasets from Darts (Herzen et al., 2022). We compare the performance of the proposed time-shift operator with conventional models such as ARIMA (Box and Jenkins, 1976) and with widely used neural networks-based models (TCN (Lea et al., 2016), N-BEATS (Oreshkin et al., 2020), N-HiTS (Challu et al., 2023)). Additionally, we compared with the non-parametric Spectral Mixture Gaussian Process (SM-GP) (Wilson and Adams, 2013) and LLMTIME (Gruver et al., 2024). We used normalized mean absolute error (NMAE)(21) as the evaluation metric. Figure 6 shows the testing errors from KRNO in comparison to other baseline methods presented in (Gruver et al., 2024). The time-shift operator is among the top 3 performing methods on 5 out of 8 datasets in the Darts collection: see Table 10 in Appendix for details.

**M4 benchmarks** The M4 dataset (Makridakis et al., 2020) is a collection of 100,000 univariate time series from diverse domains such as finance and demographics. This collection comprises six datasets corresponding to different seasonalities, varying from hourly to yearly. On this challenging dataset, the top two winning methods in the M4 competition, Smyl (2020) and Montero et al. (2020), Koopman Neural Forecaster (KNF) (Wang et al., 2022), and Nbeats-I+G (Oreshkin et al., 2020) are considered as baselines. All the models are evaluated using the symmetric mean absolute percentage error (sMAPE) metric used in the M4 competition. A comparison of KNRO performance on M4 data is presented in Table 5. We observe that KRNO is among the top two methods on datasets such as M4-Weekly and M4-Daily, where seasonality trends are not present (Wang et al., 2022).

**Crypto and Player Trajectory datasets** The Crypto (Ticchi et al., 2021) dataset is a multivariate time series containing eight features corresponding to trading prices of 14 cryptocurrencies. The objective is to forecast the returns for all 14 cryptocurrencies. The Player Trajectory dataset is a bi-variate time series containing the positions of NBA basketball players. The goal here is to forecast

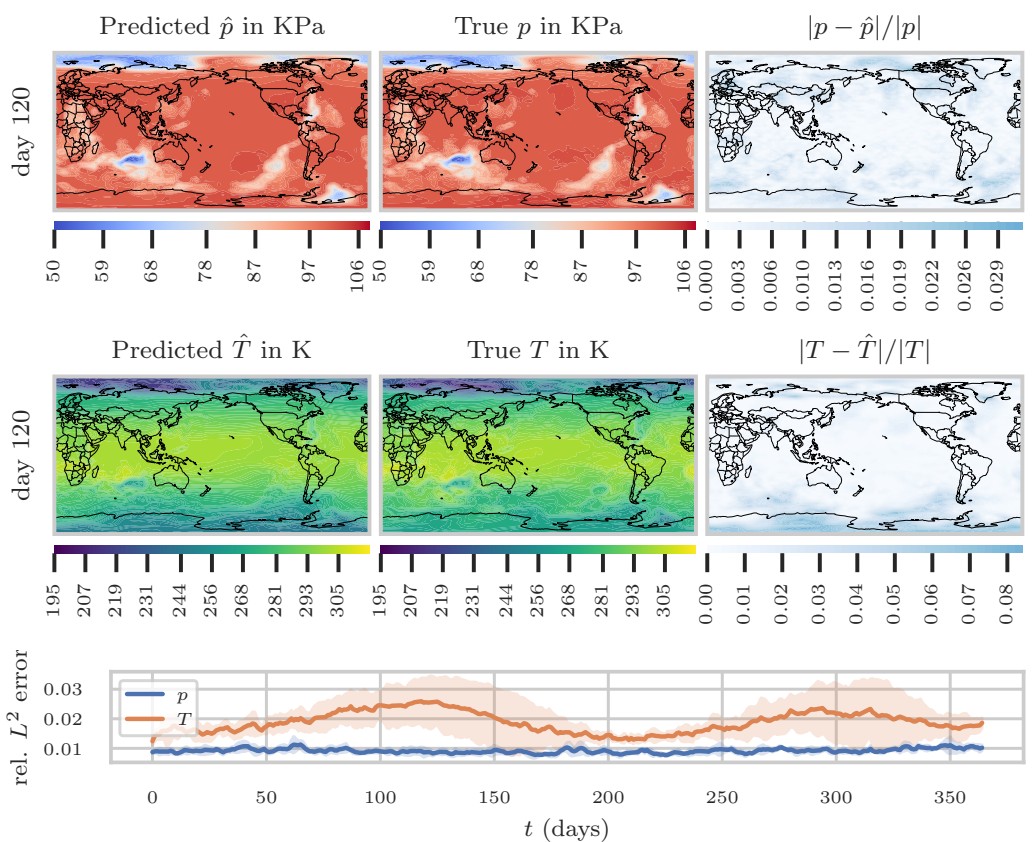

Figure 5: Climate modeling problem: Top two figures show the predicted surface pressure and temperature fields using KRNO model along with the true fields for a single day in a forecasted year. Bottom figure shows the error bars representing the $L^2$ relative errors for the five years in test data, with the shaded region indicating $\pm 1$ standard deviation.

Table 5: Comparison of sMAPE from KRNO method with other baseline methods for M4. Results with $(\cdot)^\dagger$ were taken from Wang et al. (2022).

| Method | Quarterly | Weekly | Daily |
|---|---|---|---|
| Montero et al. (2020) | 9.733 | $7.625^\dagger$ | $3.097^\dagger$ |
| Smyl (2020) | 9.679 | $7.817^\dagger$ | $3.170^\dagger$ |
| Nbeats-I+G | **9.212** | - | - |
| KNF | $10.008^\dagger$ | $7.254^\dagger$ | **$2.990^\dagger$** |
| TS-KRNO (ours) | 10.503 | **6.934** | 3.086 |

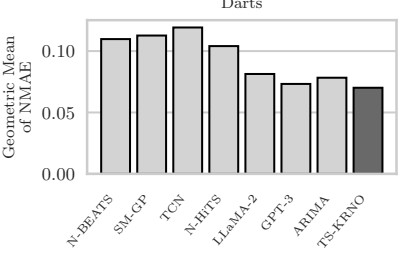

Figure 6: Comparison of geometric mean of normalized MAE on Darts datasets for various methods.

the positions of the players. For both datasets, we utilized the same training, validation, and test data as used by Wang et al. (2022). We employed weighted RMSE (Ticchi et al., 2021) for the Crypto data and RMSE for the Player Trajectory data for evaluation. Our method is compared with KNF (Wang et al., 2022) and other baseline methods such as Vector ARIMA (VARIMA) (Stock and Watson, 2001), Multi-layer Perceptron (MLP) (Faloutsos et al., 2018), FedFormer (Zhou et al., 2022), Long Expressive Memory (LEM) (Rusch et al., 2021), Variational Beam Search (VBS) (Li et al., 2021), used by Wang et al. (2022). Table 6 compares KRNO with these baseline methods. KRNO is the second-best method after KNF on Crypto and Player Trajectory datasets.

Table 6: Comparison of RMSE from KRNO method with other baseline methods on Crypto and Player Trajectory datasets.

| Model | Crypto (Weighted RMSE $10^{-3}$) | | | | Basketball Player Trajectory (RMSE) | | | |
|---|---|---|---|---|---|---|---|---|
| | (1~5) | (6~10) | (11~15) | Total | (1~10) | (11~20) | (21~30) | Total |
| VARIMA | $6.09_{\pm0.00}$ | $8.83_{\pm0.00}$ | $10.74_{\pm0.00}$ | $8.76_{\pm0.00}$ | $\mathbf{0.22}_{\pm0.00}$ | $\underline{0.90}_{\pm0.00}$ | $1.98_{\pm0.00}$ | $\mathit{1.26}_{\pm0.00}$ |
| MLP | $6.68_{\pm1.53}$ | $7.95_{\pm0.33}$ | $8.64_{\pm0.55}$ | $7.85_{\pm0.35}$ | $0.73_{\pm0.20}$ | $1.64_{\pm0.31}$ | $2.77_{\pm0.42}$ | $1.91_{\pm0.32}$ |
| MLP+RevIN+TB | $\mathbf{5.03}_{\pm0.08}$ | $\mathit{7.16}_{\pm0.13}$ | $8.41_{\pm0.06}$ | $\mathit{7.01}_{\pm0.08}$ | $0.37_{\pm0.02}$ | $1.16_{\pm0.03}$ | $2.25_{\pm0.04}$ | $1.48_{\pm0.25}$ |
| RF+TB | $6.62_{\pm1.30}$ | $7.99_{\pm0.24}$ | $8.51_{\pm1.19}$ | $7.84_{\pm0.04}$ | $0.86_{\pm0.01}$ | $2.10_{\pm0.02}$ | $3.48_{\pm0.02}$ | $2.40_{\pm0.01}$ |
| FedFormer | $5.61_{\pm0.05}$ | $7.50_{\pm0.03}$ | $8.89_{\pm0.03}$ | $7.46_{\pm0.04}$ | $0.43_{\pm0.02}$ | $\mathit{0.92}_{\pm0.02}$ | $\mathit{1.97}_{\pm0.04}$ | $1.29_{\pm0.03}$ |
| LEM | $\mathit{5.27}_{\pm0.02}$ | $7.23_{\pm0.06}$ | $\mathit{8.23}_{\pm0.05}$ | $7.02_{\pm0.04}$ | $0.33_{\pm0.01}$ | $1.08_{\pm0.02}$ | $2.18_{\pm0.02}$ | $1.42_{\pm0.02}$ |
| VBS | $15.23_{\pm0.00}$ | $14.46_{\pm0.01}$ | $26.49_{\pm0.01}$ | $19.52_{\pm0.00}$ | $0.90_{\pm0.00}$ | $2.84_{\pm0.00}$ | $9.24_{\pm0.00}$ | $5.60_{\pm0.00}$ |
| KNF | $\underline{5.24}_{\pm0.00}$ | $\mathbf{7.03}_{\pm0.01}$ | $\mathbf{7.63}_{\pm0.01}$ | $\mathbf{6.91}_{\pm0.01}$ | $\underline{0.26}_{\pm0.01}$ | $\mathbf{0.84}_{\pm0.01}$ | $\mathbf{1.81}_{\pm0.01}$ | $\mathbf{1.16}_{\pm0.01}$ |
| TS-KRNO | $\mathit{5.27}_{\pm0.27}$ | $\underline{7.07}_{\pm0.17}$ | $\underline{7.72}_{\pm0.1}$ | $\underline{6.95}_{\pm0.16}$ | $\mathit{0.27}_{\pm0.03}$ | $\mathit{0.93}_{\pm0.05}$ | $\underline{1.94}_{\pm0.07}$ | $\underline{1.25}_{\pm0.05}$ |

## 4 RELATED WORK

The nonlinear time-shift operator considered here is distinct from the Koopman operator (Koopman, 1931b), an infinite-dimensional linear operator defined over a space of observables that has been extensively studied (Wang et al., 2022; Liu et al., 2023). The time-shift operator corresponding to a set of sufficiently smooth observables can be viewed as a continuous extension of the Koopman operator (Zhen et al., 2021). This theoretical connection deserves further study.

Similar to neural ODEs (Chen et al., 2018), the present approach poses time-series forecasting in a continuous setting. However, the presented approach leads to a computationally much more efficient learning problem since no adjoint solvers are needed to calculate the loss function gradients. Also in contrast to neural ODEs, our approach can also deal with spatio-temporal problems in an elegant fashion – enabling super-resolution in both space and time.

As discussed earlier, Li et al. (2020a) examined the problem of learning a neural operator that maps the solutions of the Navier-Stokes equations over the time interval $[0, 10]$ to the solution over the interval $(10, 50]$ using FNO and discussed the ability of this strategy to provide super-resolution in both space and time. This test case can be considered as a special case of the time-shift operator formalism proposed in this work with fixed time windows. Our approach is more general since we use non-stationary kernels to parametrize the time-shift operator and treat $(t_p, t_f)$ as hyperparameters. In addition, as discussed previously the proposed KRNO model is significantly more parsimonious than FNO while providing superior accuracy.

The use of transformer-based models in time-series forecasting (Chen et al., 2021; Zhou et al., 2022; Wu et al., 2022; Liu et al., 2022; 2024) and neural operators incorporating the attention mechanism (Kissas et al., 2022) is becoming increasingly popular. It is worth noting that transformer-based models developed for time-series forecasting assume that the observations are regularly sampled; see, for example, FedFormer (Zhou et al., 2022) used in our benchmarking studies.

## 5 CONCLUSION

In this work, we showed how the time-shift operator can be learned in a supervised setting for temporal and spatio-temporal problems. We proposed a novel Khatri-Rao neural operator that enables a highly flexible yet parsimonious parametrization of the time-shift operator. Numerical studies on a range of benchmark problems suggest that the proposed approach compares favourably against state-of-the-art methods in time-series forecasting and neural operator learning. Our method achieved top performance on 8/27 test cases and top-3 performance on 19/27 test cases. We hope that the present work will enable further advances in operator-theoretic frameworks for time-series forecasting and spatio-temporal modeling.

REPRODUCIBILITY

The KRNO architecture codebase and the scripts used to generate the results is available at `https://anonymous.4open.science/r/KRNO/README.md`. Further information about the experiments and the datasets used are provided in Section 3 and in the Appendix.

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

APPENDIX

## A  CONTINUITY OF THE TIME-SHIFT OPERATOR

**Lemma 1.** If $F : \mathbb{R}^n \times \mathbb{R} \to \mathbb{R}^n$ is Lipschitz continuous over $[t_p, t_f]$, then $\exists C > 0$ such that

$$||A_{t_p}^{t,t_f} z_1 - A_{t_p}^{t,t_f} z_2||_{L^2((t,t_f];\mathbb{R}^n)} \leq C||z_1 - z_2||_{L^2([t_p,t];\mathbb{R}^n)},$$

where $t_p < t < t_f$.

*Proof.* Let $z_1, z_2 \in L^2([t_p, t_f]; \mathbb{R}^n)$ denote trajectories corresponding to two different initial conditions and let $e = z_1 - z_2$. Then, we have

$$||A_{t_p}^{t,t_f} z_1 - A_{t_p}^{t,t_f} z_2||_{L^2((t,t_f];\mathbb{R}^n)} = ||z_1 - z_2||_{L^2((t,t_f];\mathbb{R}^n)} = ||e||_{L^2((t,t_f];\mathbb{R}^n)}. \tag{9}$$

Noting that $||\dot{e}||_2 = ||F(z_1) - F(z_2)||_2 \leq L_F ||e||_2$, where $L_F$ is the Lipschitz constant of $F$, we have

$$\frac{d}{d\tau} ||e(\tau)||_2^2 = 2e(\tau)^T \frac{de}{d\tau} \leq 2 ||e(\tau)||_2 \left\|\frac{de}{d\tau}\right\|_2 = 2L_F ||e(\tau)||_2^2, \ \tau \in [t_p, t_f]. \tag{10}$$

Applying Grönwall's lemma (Ames and Pachpatte, 1997) to the preceding inequality, we have

$$||e||_{L^2((t,t_f];\mathbb{R}^n)}^2 = \int_t^{t_f} ||e(\tau)||_2^2 d\tau \leq ||e(t_p)||_2^2 \int_t^{t_f} e^{2L_F(\tau - t_p)} d\tau. \tag{11}$$

Since the trajectories are continuous over $[t_p, t_f]$, it follows from the extreme value theorem that $||F(z(t))||_2$ is bounded over this interval which in turn implies that $\dot{z} \in L^\infty([t_p, t_f]; \mathbb{R}^n) \subset L^2([t_p, t_f]; \mathbb{R}^n)$. In addition, since $z$ is square integrable $z \in H^1([t_p, t_f]; \mathbb{R}^n)$. Noting that $H^1([t_p, t_f]; \mathbb{R}^n) \hookrightarrow C([t_p, t_f]; \mathbb{R}^n)$ due to the Sobolev embedding theorem, there exists an embedding constant $C_1 > 0$ such that

$$\begin{aligned} ||e(t_p)||_2 &\leq ||e||_{L^\infty([t_p,t];\mathbb{R}^n)} \leq C_1 \left( ||e||_{L^2([t_p,t];\mathbb{R}^n)} + ||\dot{e}||_{L^2([t_p,t];\mathbb{R}^n)} \right) \\ &= C_1(1 + L_F) ||e||_{L^2([t_p,t];\mathbb{R}^n)}. \end{aligned} \tag{12}$$

Using (9), (11), (12), we have

$$||A_{t_p}^{t,t_f} z_1 - A_{t_p}^{t,t_f} z_2||_{L^2((t,t_f];\mathbb{R}^n)} \leq C||z_1 - z_2||_{L^2([t_p,t];\mathbb{R}^n)},$$

where $C = C_1(1 + L_F)(2L_F)^{-0.5} \sqrt{\exp(2L_F(t_f - t_p)) - \exp(2L_F(t - t_p))}$.

$\square$

The continuity of the spatio-temporal time-shift operator can be established for time-dependent PDEs under appropriate regularity assumptions. We leave this for future work.

## B  PROOF FOR PROPOSITION 1

We start by briefly clarifying what is meant by the quadrature nodes lying on a product grid. An example of a two-dimensional product grid is provided in Figure 7 below.

Proposition 1 states that if:

1. the quadrature nodes lie on a product grid, $X = \bar{t} \times x^{(1)} \times \ldots x^{(d-1)}$ where $\bar{t}, x^{(i)} \in \mathbb{R}^n$ indicates the quadrature nodes along the time dimension and the $i^{\text{th}}$ dimension of $x$ respectively such that $N = n^d$ (for general case $N = \Pi_{i=1}^d n_i$) and

2. the kernel function has a component-wise product structure of the form given in Equation (7) reproduced below for clarity:

$$\kappa(\{t, x\}, \{t', x'\}) = \kappa^{(1)}(t, t') \odot \left( \odot_{i=2}^d \kappa^{(i)}([x]_{i-1}, [x]'_{i-1}) \right)$$

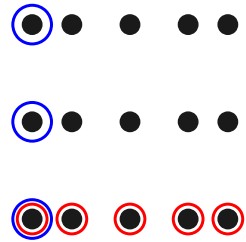

Figure 7: Example of a quadrature rule which lies on a product grid. Circled in blue are the quadrature nodes along the first dimension and circled in red are the quadrature nodes along the second dimension.

Then the kernel function evaluated at the quadrature nodes inherits the Khatri-Rao product structure provided in Equation (8) and reproduced below,

$$\kappa(X, X) = \kappa^{(1)}(\bar{t}, \bar{t}) * \left( \underset{i=2}{\overset{d}{*}} k^{(i)}(x^{(i-1)}, x^{(i-1)}) \right).$$

Here $\kappa^{(i)}(\cdot, \cdot) \in \mathbb{R}^{qn \times pn}$ is a block-partitioned matrix where block $jk$ is the $jk^{\text{th}}$ output from the component kernel $\kappa^{(i)}$ evaluated on the outer product of the quadrature nodes along the $i^{\text{th}}$ dimension.

*Proof.* We start by observing that $\kappa(X, X)$ can be block-partitioned into $q \times p$ blocks of size $N \times N$.

$$\kappa(X, X) = \begin{bmatrix} \kappa_{1,1} & \kappa_{1,2} & \dots & \kappa_{1,p} \\ \kappa_{2,1} & \kappa_{2,2} & \dots & \kappa_{2,p} \\ \vdots & & \ddots & \vdots \\ \kappa_{q,1} & \kappa_{q,2} & \dots & \kappa_{q,p} \end{bmatrix}. \tag{13}$$

Each of these $N \times N$ blocks inherit the product structure of Equation (8),

$$\kappa_{j,k} = \odot_{i=1}^{d} \kappa_{j,k}^{(i)}(X[:, i-1], X[:, i-1]), \tag{14}$$

where $\kappa_{j,k}^{(i)}(X[:, i-1], X[:, i-1]) \in \mathbb{R}^{N \times N}$ is the $jk^{\text{th}}$ output of the $i^{\text{th}}$ component kernel function evaluated on the $i^{\text{th}}$ dimension of the quadrature nodes. Following Saatçi (2011), we can write the $jk^{\text{th}}$ block in the kernel evaluated at the quadrature nodes as the Kronecker product,

$$\kappa_{j,k} = \kappa_{j,k}^{(1)}(\bar{t}, \bar{t}) \otimes \left( \otimes_{i=2}^{d} \kappa_{j,k}^{(i)}(x^{(i-1)}, x^{(i-1)}) \right), \tag{15}$$

where $\kappa_{j,k}^{(i)}(x^{(i-1)}, x^{(i-1)}) \in \mathbb{R}^{n \times n}$, each $\bar{t}, x^{(i)} \in \mathbb{R}^{n}$ indicates the one-dimensional quadrature nodes along the time and $i^{\text{th}}$ dimension respectively. The Khatri-Rao product structure follows from substituting (15) into (13). $\qquad \square$

The above proof can be generalized to cases where the quadrature nodes are distributed on a product grid with a variable number of nodes along each dimension, i.e., $N = \Pi_{i=1}^{d} n_i$.

## C  GENERALIZATION TO DIFFERENT INPUT AND OUTPUT DOMAINS

In this section, we generalize Proposition 1 to the case where the input and output are defined over different spatio-temporal domains and different quadrature nodes are used for the input and output functions. We start by rewriting the kernel integral transform layer as a map with the input function $v_\ell : \Omega_\ell \times \mathcal{I}_\ell \to \mathbb{R}^p$ and the output function $v_{\ell+1} : \Omega_{\ell+1} \times \mathcal{I}_{\ell+1} \to \mathbb{R}^q$ as follows

$$v_{\ell+1}(t_{\ell+1}, x_{\ell+1}) = \mathcal{K}(v_\ell)(t_{\ell+1}, x_{\ell+1})$$
$$= \int_{\Omega_\ell} \int_{(0,\tau]} \kappa(\{t_{\ell+1}, x_{\ell+1}\}, \{t'_\ell, x'_\ell\}) v_\ell(t'_\ell, x'_\ell) dt'_\ell dx'_\ell + W v_\ell(\Phi_\ell(t_{\ell+1}, x_{\ell+1})) + b,$$

where $\kappa : \mathbb{R} \times \Omega_\ell \times \mathbb{R} \times \Omega_{\ell+1} \to \mathbb{R}^{q \times p}$ is a matrix-valued kernel, $W \in \mathbb{R}^{q \times p}$ is a weight matrix, $\Phi_\ell : \Omega_{\ell+1} \to \Omega_\ell$ is a map between the output and input domains, and $b \in \mathbb{R}^q$ is a bias vector.

Let $X_\ell$ and $X_{\ell+1}$ denote the sets of quadrature nodes for the input and output domains, respectively. The quadrature nodes over the domain of the input function are assumed to lie on a product grid, i.e., $X_\ell = \bar{t}_\ell \times x_\ell^{(1)} \times \ldots x_\ell^{(d-1)}$, where $\bar{t}_\ell \in \mathbb{R}^{n_\ell}$ and $x_\ell^{(i)} \in \mathbb{R}^{n_\ell}$ denote the quadrature nodes along the input time dimension and the $i^{\text{th}}$ dimension of $x_\ell$, respectively, such that $N_\ell = n_\ell^d$ (for the general case when the number of quadrature nodes along the $i^{\text{th}}$ dimension is $n_{\ell_i}$, we have $N_\ell = \Pi_{i=1}^d n_{\ell_i}$). Similarly, the quadrature nodes over the output domain are assumed to lie on a product grid, $X_{\ell+1} = \bar{t}_{\ell+1} \times x_{\ell+1}^{(1)} \times \ldots x_{\ell+1}^{(d-1)}$ where $\bar{t}_{\ell+1} \in \mathbb{R}^{n_{\ell+1}}$ and $x_{\ell+1}^{(i)} \in \mathbb{R}^{n_{\ell+1}}$ denote the quadrature nodes along the output time dimension and the $i^{\text{th}}$ dimension of $x_{\ell+1}$, respectively, such that $N_{\ell+1} = n_{\ell+1}^d$ (for the general case with different number of quadrature nodes along each dimension $N_{\ell+1} = \Pi_{i=1}^d n_{\ell+1_i}$). As before, we will consider a kernel with a component-wise product structure of the form given in Equation (7).

Similar to the previous proof, we start by observing that $\kappa(X_{\ell+1}, X_\ell)$ can be block-partitioned into $q \times p$ blocks of size $N_{\ell+1} \times N_\ell$, i.e.,

$$\kappa(X_{\ell+1}, X_\ell) = \begin{bmatrix} \kappa_{1,1} & \kappa_{1,2} & \ldots & \kappa_{1,p} \\ \kappa_{2,1} & \kappa_{2,2} & \ldots & \kappa_{2,p} \\ \vdots & & \ddots & \vdots \\ \kappa_{q,1} & \kappa_{q,2} & \ldots & \kappa_{q,p} \end{bmatrix}. \tag{16}$$

Each $N_\ell \times N_{\ell+1}$ block inherits the product structure, i.e., $\kappa_{j,k} = \odot_{i=1}^d \kappa_{j,k}^{(i)}(X_{\ell+1}[:, i-1], X_\ell[:, i-1])$, where $\kappa_{j,k}^{(i)}(X_{\ell+1}[:, i-1], X_\ell[:, i-1]) \in \mathbb{R}^{N_{\ell+1} \times N_\ell}$ is the $jk^{\text{th}}$ output of the $i^{\text{th}}$ component kernel function evaluated on the $i^{\text{th}}$ dimension of the quadrature nodes. The $jk^{\text{th}}$ block in the kernel evaluated at the quadrature nodes can be written as

$$\kappa_{j,k} = \kappa_{j,k}^{(1)}(\bar{t}_{\ell+1}, \bar{t}_\ell) \otimes \left( \otimes_{i=2}^d \kappa_{j,k}^{(i)}(x_{\ell+1}^{(i-1)}, x_\ell^{(i-1)}) \right), \tag{17}$$

where $\kappa_{j,k}^{(i)}(x_{\ell+1}^{(i-1)}, x_\ell^{(i-1)}) \in \mathbb{R}^{n_{\ell+1} \times n_\ell}$. Substituting (17) into (16), we have

$$\kappa(X_{\ell+1}, X_\ell) = \kappa^{(1)}(\bar{t}_{\ell+1}, \bar{t}_\ell) * \left( \underset{i=2}{\overset{d}{*}} k^{(i)}(x_{\ell+1}^{(i-1)}, x_\ell^{(i-1)}) \right), \tag{18}$$

where $\kappa^{(i)}(\cdot, \cdot) \in \mathbb{R}^{qn_{\ell+1} \times pn_\ell}$ is a block-partitioned matrix where block $jk$ is the $jk^{\text{th}}$ output from the component kernel $\kappa^{(i)}$ evaluated on the outer product of the quadrature nodes along the $i^{\text{th}}$ dimension.

It follows from this result that we retain the original computational complexity of the KRNO operator even in situations where the inputs and outputs are defined over different domains. In addition, this result provides the flexibility of designing memory-efficient multi-resolution neural operators, where the hidden layers operate on variable-resolution representations of the input function. This generalized KRNO integral transform layer can be viewed as a continuous analog of upsampling and downsampling layers used in convolutional neural networks.

## D   ALGORITHM FOR KHATRI-RAO STRUCTURED MATRIX-VECTOR PRODUCTS

In this section, we present an algorithm to efficiently compute the matrix-vector product associated with the Khatri-Rao product structured matrix defined in (8), without the need to explicitly construct the full matrix of size $qN \times pN$.

Let $A \in \mathbb{R}^{qN \times pN}$ be a block structured matrix of the form,

$$A = \begin{bmatrix} A_{1,1} & A_{1,2} & \ldots & A_{1,p} \\ A_{2,1} & A_{2,2} & \ldots & A_{2,p} \\ \vdots & & \ddots & \vdots \\ A_{q,1} & A_{q,2} & \ldots & A_{q,p} \end{bmatrix} \tag{19}$$

where each $A_{j,k} = \otimes_{i=1}^{d} A_{j,k}^{(i)}$ and $A_{j,k}^{(i)} \in \mathbb{R}^{n \times n}$. Assuming $q, p << N$, the computational complexity associated with the matrix-vector product $u = Av$ can be reduced from $\mathcal{O}(N^2)$ to $\mathcal{O}(N^{1+1/d})$. In addition, the memory requirements are also reduced from $\mathcal{O}(N^2)$ to $\mathcal{O}(N^{2/d} + N)$. An efficient PyTorch implementation outlining the steps is provided below.

```python
def khatri_rao_mmprod(
    A: list[Float[Tensor, "q p n1 n2"]], V: Float[Tensor, "pN batch"]
) -> Float[Tensor, "qN batch"]:
    d = len(A) # size of the product grid (# of kernel components)
    q, p, _, _ = A[0].shape
    pN, bs = V.shape
    X = V.reshape(p, -1, bs).transpose(-2, -1)
    for i in range(d):
        Gd = A[i].shape[-1]
        bs_prod = X.shape[:-1]
        X = X.reshape(*bs_prod, Gd, -1)
        Z = A[i].unsqueeze(-3) @ X
        X = Z.transpose(-2, -1).reshape(q, p, bs, -1)
    return X.sum(1).transpose(-2, -1).reshape(-1, bs)
```

We note that the above algorithm is applicable to Khatri-Rao product structured matrix, as defined in (18), where the inputs and outputs are defined over different spatio-temporal domains (with each domain using a different set of quadrature nodes).

## E    DETAILS ON KRNO PARAMETRIZATION

As is mentioned in the paper, we parametrize each component-wise kernel, $\kappa^{(i)} : \mathbb{R} \times \mathbb{R} \to \mathbb{R}^{q \times p}$ by a neural network. All neural nets use skip connections and layer normalization (Ba et al., 2016). In addition, before passing an input into the component function, we apply an input transformation $\phi : \mathbb{R} \times \mathbb{R} \to \mathbb{R}^m$,

$$\phi(t, t') = \frac{1}{\sqrt{2}} \cos\left([t \quad t']\,\omega + \beta\right), \tag{20}$$

where $\omega \in \mathbb{R}^{2 \times m}$ and $\beta \in \mathbb{R}^m$. Such input feature transforms were found to be beneficial in prior works (Kissas et al., 2022)

## F    KRNO PERFORMANCE ON IRREGULARLY SPACED TIME SERIES DATA

This section presents additional numerical results to evaluate the performance of KRNO on irregularly spaced time-series data obtained for the two-dimensional spiral test case from Chen et al. (2018). We consider two sets of 10 irregularly spaced training trajectories, which represent moderate and high levels of irregularity, alongside one equispaced training trajectory. The equispaced trajectory is generated by sampling the states at 100 evenly spaced time stamps over the interval $[0, 15.61]$ seconds. To generate training datasets with a moderate level of irregularity, 100 new time-stamps are obtained by adding random noise $\epsilon_t \sim \mathcal{U}[0, 0.078]$ to the time stamps of the equispaced trajectory. Subsequently, the training trajectory is obtained by sampling the states at the randomly perturbed time-stamps. This randomization procedure was repeated to create 10 different training datasets. Similarly, time stamps for the training trajectories with a high degree of irregularity are obtained by adding random noise $\epsilon_t \sim \mathcal{U}[0, 0.156]$ to the time stamps of the equispaced trajectory. This process was again repeated to create 10 different training datasets where the distribution of the time-stamps are highly irregular making this test-case more challenging. We trained 21 KRNO models in total and the performance of the models are evaluated on a test dataset containing the states at 20 uniformly spaced time stamps over the interval $(15.61, 18.76]$ seconds.

The performance of the KRNO model trained on the equispaced trajectory was compared with models trained on the two sets of irregularly spaced trajectories. For all models, the time-shift operator was trained by fixing the length of the time window corresponding to the input and the output to 1.419 seconds. We didn't tune the KRNO hyperparameters for this experiment. The KRNO architecture

consisted of 128 channels in both the lifting and projection layers, with three kernel integral layers, each containing four channels.

The predictive performance of the KRNO models are compared in Table 7. For the case of irregularly-spaced training data, we provide the statistics of test RMSE and MSE over 10 different randomly sampled trajectories generated using the procedure described earlier. It can be noted from Table 7 that KRNO models trained on trajectories with a moderate level of irregularity demonstrates an average predictive performance that is close to the model trained on an equispaced trajectory. The average test RMSE and MSE error are higher for models trained on trajectories with a high degree of irregularity.

Figure 8 compares the predictions made by the KRNO model trained on the equispaced data to representative predictions made by two of the models trained on randomly sampled trajectories with low and high irregularity. It can be seen that the predictions made by the KRNO model trained on the trajectory with a moderate level of irregularity is close to the the model trained on the equispaced trajectory. We note that reduction in predictive accuracy when the distribution of the time-stamps are highly irregular is influenced by the time intervals where the sampling frequency is low. For instance, lower sampling frequency in the time interval close to the testing time window has the most significant impact on predictive accuracy. In such situations, the hyperparameters $t_p$ and $t_f$ would need to carefully selected to improve the predictive performance.

Another important factor that impacts the predictive accuracy for irregularly-spaced training observations is the quadrature scheme used to approximate the kernel integral transform layers. Our current implementation uses a trapezoidal quadrature rule which is not ideal when the training data is sampled at a highly irregular frequency. It is expected that by adopting a quadrature scheme that obtains weights on-the-fly for irregularly spaced data (while meeting a target precision), the accuracy can be improved further.

Table 7: Comparison of test RMSE and MSE of KRNO models trained on equispaced trajectory and trajectories with moderate and high levels of irregularity in the distribution of the time-stamps. For irregularly-spaced time-series data, we provide mean and standard deviation of test errors (RMSE and MSE) for models trained over 10 different randomly sampled trajectories.

| Train trajectory type | RMSE | MSE |
|:---:|:---:|:---:|
| Equispaced | 0.188 | 0.035 |
| Moderate irregularity | $0.195 \pm 0.05$ | $0.040 \pm 0.02$ |
| High irregularity | $0.249 \pm 0.05$ | $0.064 \pm 0.03$ |

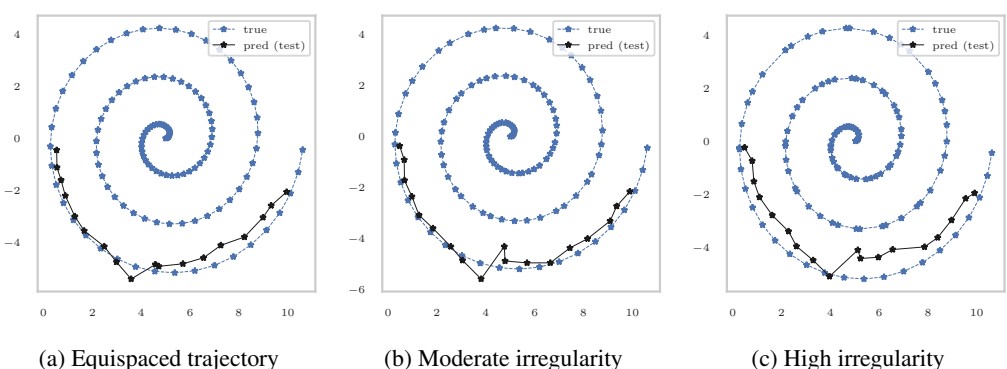

(a) Equispaced trajectory        (b) Moderate irregularity        (c) High irregularity

Figure 8: Left figure shows the predictions of KRNO model trained on equispaced trajectory. Middle and right figures show representative predictions of KRNO models trained on trajectories with moderate and high levels of irregularity in the distribution of time-stamps, respectively.

## G  RESOURCE COMPARISON BETWEEN KRNO AND FNO FOR DIFFERENT SPATIAL RESOLUTIONS

In this section, we compare the resource usage between KRNO and FNO-3D for different spatial resolutions using the shallow water dataset. We utilized the default settings for KRNO as mentioned

in the shallow water problem, except for the quadrature grid. For all the spatial resolutions, we employed a quadrature grid of size $32 \times 32 \times 5$ in the hidden KRNO integral transform layers. It is important to note that for high spatial resolution data, resource usage is primarily driven by computations in the first and last integral transform layers, which contain the highest number of quadrature nodes. For the FNO-3D model, the number of Fourier modes in each spatial dimension is set to the default value of 12. Increasing this value to 64 results in a substantial increase in memory usage (10,324 MB) and training time per iteration (0.1725 seconds) for a spatial resolution of $160 \times 160$. This also increases the parameter count of FNO-3D from 5.5 million to 157 million. In contrast, KRNO maintains a fixed parameter count of 145319, independent of the size of the quadrature grid and the resolution of the dataset.

The results for resource usage during training and inference are shown in Table 8, Table 9, and Figure 9. It can be noted that even though we use a high-resolution quadrature grid for the first and the last layers in this numerical study, the time complexity of KRNO is comparable to that of FNO-3D when the number of Fourier modes are set to 12. As discussed earlier, the memory requirements and time complexity of FNO-3D will increase dramatically with increase in the number of Fourier modes. We observe that the memory usage of KRNO is significantly higher during training when compared to inference. We believe that this can be reduced by further optimizing the implementation of Khatri-Rao matrix-vector products.

We would like to highlight that our current implementation uses a mid-point quadrature scheme for the temporal dimension and a trapezoidal quadrature scheme for spatial dimensions to evaluate the integral transform layers. Additionally, in our current implementation, the quadrature nodes are defined over the spatial mesh of the input function. While this approach is reasonable for the problems we are considering, it is not the most suitable (and tends to be overly conservative) for high-resolution spatio-temporal datasets. For such datasets, the memory requirements can be significantly reduced by using quadrature nodes that are defined over a lower-resolution spatial grid which is independent of the input's spatial resolution.

Alternatively, we could design a quadrature scheme that uses a low-resolution subsampling of the input as nodes and generates quadrature weights on-the-fly to meet a specified target precision. This would not only enhance accuracy and efficiency for high-resolution spatio-temporal datasets but also improve the performance of KRNO for irregularly spaced observations. We plan to explore this in future work.

Table 8: Resource usage while training KRNO and FNO-3D models on shallow water problem for different spatial resolutions for a batch size of 8 training points. Note that the number of Fourier modes for FNO-3D is set to 12.

| Spatial resolution | # Quadrature nodes | GPU memory (MB) | | Time (seconds) | |
|---|---|---|---|---|---|
| | | KRNO | FNO-3D | KRNO | FNO-3D |
| $32 \times 32$ | 5,120 ($N$) | 1,390 | 854 | 0.0279 | 0.0216 |
| $64 \times 64$ | 20,480 ($4N$) | 2,776 | 1,350 | 0.0394 | 0.0252 |
| $96 \times 96$ | 46,080 ($9N$) | 4,884 | 2,124 | 0.0626 | 0.0405 |
| $128 \times 128$ | 81,920 ($16N$) | 7,608 | 3,318 | 0.0999 | 0.0704 |
| $160 \times 160$ | 128,000 ($25N$) | 10,040 | 4,872 | 0.1584 | 0.1114 |

**Note:** $N = 32 \times 32 \times 5 = 5120$.

Table 9: Resource usage during inferencing KRNO and FNO-3D models on shallow water problem for different spatial resolutions for a batch size of 8 test points. Note that the number of Fourier modes for FNO-3D is set to 12.

| Spatial resolution | # Quadrature nodes | GPU memory (MB) | | Time (seconds) | |
|---|---|---|---|---|---|
| | | KRNO | FNO-3D | KRNO | FNO-3D |
| $32 \times 32$ | 5,120 ($N$) | 708 | 942 | 0.0107 | 0.0065 |
| $64 \times 64$ | 20,480 ($4N$) | 1,314 | 1,294 | 0.0134 | 0.0069 |
| $96 \times 96$ | 46,080 ($9N$) | 2,366 | 1,622 | 0.0185 | 0.0108 |
| $128 \times 128$ | 81,920 ($16N$) | 3,796 | 2,428 | 0.0312 | 0.0210 |
| $160 \times 160$ | 128,000 ($25N$) | 5,644 | 3,292 | 0.0502 | 0.0315 |

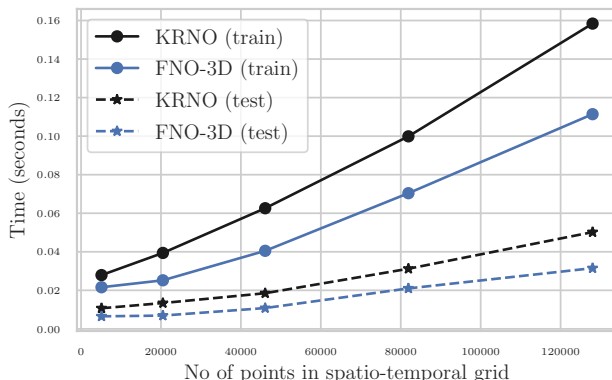

Figure 9: Shallow water problem: Comparison of training and testing times per iteration for KRNO and FNO-3D models as a function of the number of quadrature nodes. For the FNO-3D model, the number of Fourier modes in each spatial dimension is set to 12. Note that increasing the number of Fourier modes to 64 results in a factor of 30 increase in parameter count and the training/inference times exceeds the KRNO model; see the discussion in Appendix G for more details.

## H ADDITIONAL EXPERIMENTAL DETAILS

This section provides additional details on the experimental setup used to generate the results presented in the main text. For all the numerical studies, the KRNO network architecture had 128 channels in both the lifting and projection layers and three kernel integral layers. In each test case, we used a same input and output quadrature grids in each kernel integral layer. For problems involving high spatial or temporal resolutions, adopting lower-resolution quadrature grids within the internal kernel integral layers is recommended as an effective strategy to reduce computational costs. As mentioned in the section E, the kernel function used in each integral transform layer is parameterized by a neural network containing three hidden layers. Additional hyperparameters used in hyperparameter tuning for Darts, M4, Cryto, and Player Trajectory datasets are summarized in Table 11 and Table 13, respectively. AdamW (Loshchilov and Hutter, 2017) optimizer is used for training all the models. All the computations were carried out on a single Nvidia RTX 4090 with 24GB memory.

In all experiments, we treat the $t_p$ and $t_f$ as fixed hyperparameters. We would like to mention here that further work is needed to explore the possibility of training a single model on a dataset containing input/output trajectories for different settings of $t_p$ and $t_f$. This would enable the possibility of learning a model that can predict the dynamics at different lengthscales.

### H.0.1 DARTS BENCHMARKS

For all the datasets in Darts, we used 60%-20%-20% as a train-validation-test split. We performed a grid search on the Darts datasets using the hyperparameters listed in Table 11 to find the optimal hyperparameters. Model selection was done based on the NMAE on the validation set. Since the available training data in the Dart dataset is not sufficient to train a deep network, we conducted weight decay tuning to determine the optimal weight decay value using the optimal hyperparameters. This optimal weight decay value was then used to train the final model using both the train and validation data. This final model is used to get predictions by forecasting recursively until the end of the testing window, shown in Figure 10. The evaluation metric, normalized MAE (NMAE), is computed as follows

$$NMAE(\mathbf{y}, \hat{\mathbf{y}}) = \frac{MAE(\mathbf{y}, \hat{\mathbf{y}})}{\frac{1}{n} \sum_{i=1}^{n} |y_i|} = \frac{MAE(\mathbf{y}, \hat{\mathbf{y}})}{\text{mean}(|\mathbf{y}|)}, \tag{21}$$

where $\mathbf{y}$ and $\hat{\mathbf{y}}$ are the truth and predicted time series.

Table 10: Table showing top five models for each test case.

| Dataset | Test case | Best | Second | Third | Fourth | Fifth |
|---|---|---|---|---|---|---|
| Darcy flow | u | **KRNO** | FNO | POD-DeepONet | DeepONet | - |
| Hyper-elastic | σ | **KRNO** | FNO | DeepONet | - | - |
| Shallow water | $\rho$ | **TS-KRNO** | FNO-3D | LOCA | - | - |
| | $v_1$ | **TS-KRNO** | FNO-3D | LOCA | - | - |
| | $v_2$ | **TS-KRNO** | FNO-3D | LOCA | - | - |
| Darts | AirPassengers | LLaMA-2 | ARIMA | **TS-KRNO** | GPT-3 | SM-GP |
| | AusBeer | N-BEATS | LLaMA-2 | GPT-3 | **TS-KRNO** | ARIMA |
| | GasRateCO2 | SM-GP | **TS-KRNO** | ARIMA | LLaMA-2 | N-BEATS |
| | MonthlyMilk | GPT-3 | LLaMA-2 | **TS-KRNO** | SM-GP | N-HiTS |
| | sunspots | **TS-KRNO** | ARIMA | GPT-3 | LLaMA-2 | N-HiTS |
| | Wine | **TS-KRNO** | GPT-3 | ARIMA | TCN | N-HiTS |
| | Wooly | N-HiTS | ARIMA | SM-GP | **TS-KRNO** | GPT-3 |
| | HeartRate | TCN | GPT-3 | SM-GP | **TS-KRNO** | N-HiTS |
| M4 | Monthly | KNF | Nbeats-I+G | Smyl | Montero et al | **TS-KRNO** |
| | Weekly | **TS-KRNO** | KNF | Montero et al | Smyl | - |
| | Daily | KNF | **TS-KRNO** | Montero et al | Smyl | - |
| | Hourly | Smyl | KNF | Montero et al | **TS-KRNO** | - |
| | Yearly | Nbeats-I+G | Smyl | Montero et al | KNF | **TS-KRNO** |
| | Quarterly | Nbeats-I+G | Smyl | Montero et al | KNF | **TS-KRNO** |
| Crypto | $(1 \sim 5)$ | MLP+RevIN+TB | KNF | **TS-KRNO** | LEM | VARIMA |
| | $(6 \sim 10)$ | KNF | **TS-KRNO** | MLP+RevIN+TB | LEM | FedFormer |
| | $(11 \sim 15)$ | KNF | **TS-KRNO** | LEM | MLP+RevIN+TB | RF+TB |
| | Total | KNF | **TS-KRNO** | MLP+RevIN+TB | LEM | FedFormer |
| Player Traj | $(1 \sim 10)$ | VARIMA | KNF | **TS-KRNO** | LEM | MLP+RevIN+TB |
| | $(11 \sim 20)$ | KNF | VARIMA | FedFormer | **TS-KRNO** | MLP+RevIN+TB |
| | $(21 \sim 30)$ | KNF | **TS-KRNO** | FedFormer | VARIMA | LEM |
| | Total | KNF | **TS-KRNO** | VARIMA | FedFormer | LEM |

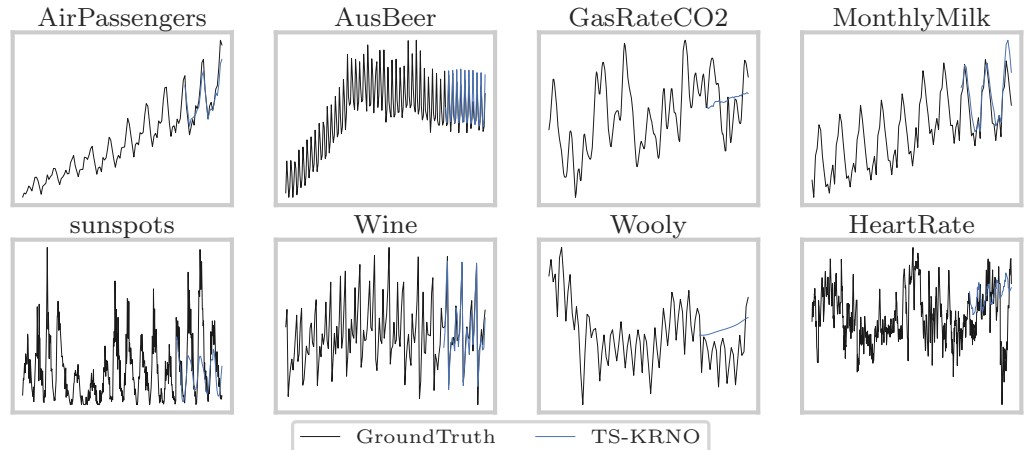

Figure 10: TS-KRNO predictions on Darts datasets

### H.0.2 M4 BENCHMARKS

We utilized the train and test datasets from the M4 competition (Makridakis et al., 2020). For all M4 datasets, the last 10% of the data for each time series in the training data is used as validation data. The testing process involves forecasting for a specified time period (testing window length) for each seasonality. The testing window lengths for each seasonality are shown in the parenthesis next to the seasonality in Table 12.

Table 11: Hyperparameter tuning ranges used for Darts dataset.

| Learning rate | Integral layer channels | Hidden units in kernel | Look-back window length | Prediction window length | ReVIN |
|---|---|---|---|---|---|
| [1e-3, 5e-3] | [5, 10, 32] | [32, 64] | 10 to 100 | 5 to 100 | [True, False] |

Table 12: Comparison of sMAPE from TSO method with other baseline methods on M4 datasets. Results with $(\cdot)^{\dagger}$ were taken from Wang et al. (2022).

| Method | Monthly(18) | Weekly(13) | Daily(14) | Hourly(48) | Yearly(6) | Quarterly(8) |
|---|---|---|---|---|---|---|
| Montero et al. (2020) | 12.639 | $7.625^{\dagger}$ | $3.097^{\dagger}$ | 11.506 | 13.528 | 9.733 |
| Smyl (2020) | 12.126 | $7.817^{\dagger}$ | $3.170^{\dagger}$ | **9.328** | 13.176 | 9.679 |
| Nbeats-I+G | 12.024 | - | - | - | **12.924** | **9.212** |
| KNF (Wang et al., 2022) | $\mathbf{11.930}^{\dagger}$ | $7.254^{\dagger}$ | $\mathbf{2.990}^{\dagger}$ | 11.294 | 13.800 | 10.008 |
| TS-KRNO(ours) | 13.432 | **6.934** | 3.086 | 11.686 | 14.302 | 10.503 |

### H.0.3 CRYPTO AND PLAYER TRAJECTORY BENCHMARKS

For these two datasets, we used the same train-test splits used by Wang et al. (2022). Similar to M4 datasets, 10% of the data corresponding to each time series in train data is used as validation data. For Crypto and Player Trajectory datasets, the testing window lengths are set to 15 and 30 as in (Wang et al., 2022).

Table 13: Hyperparameter ranges used in M4, Crypto, and Player Trajectory datasets.

| Learning rate | Integral layer channels | Hidden units in kernel | Look-back window length | Prediction window length | ReVIN |
|---|---|---|---|---|---|
| [1e-3, 5e-3] | [16, 32] | [32, 64] | 3 to 192 | 1 to 18 | [True, False] |

### H.0.4 SPATIAL MODELING PROBLEMS

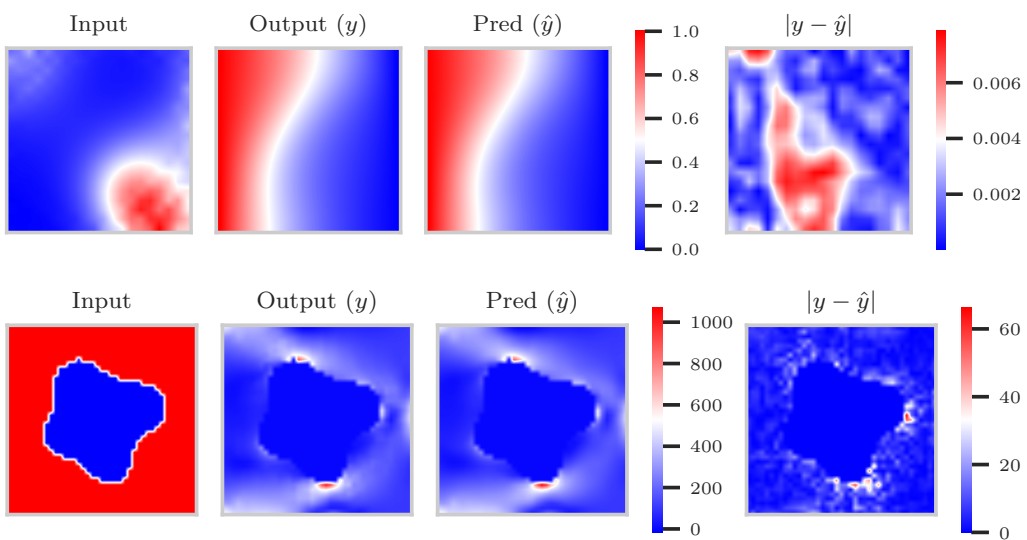

Figure 11: The top row presents a sample prediction from the test set for the Darcy-flow problem, while the bottom row illustrates a sample prediction for the elasticity problem.

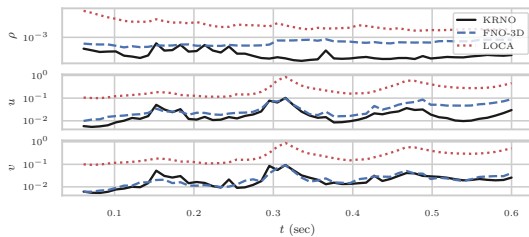

Table 14: Comparison of the average relative $L^2$ errors on the shallow water problem for the three field variables when training is conducted for 200 epochs.

| Method | $L^2$ relative error | | |
|--------|--------|--------|--------|
| | $\rho$ | $u$ | $v$ |
| FNO-3D | 0.000719 | 0.01951 | **0.01174** |
| LOCA | 0.003091 | 0.15179 | 0.14942 |
| KRNO | **0.000331** | **0.01339** | 0.01406 |

Figure 12: Comparison of the average relative $L^2$ errors as a function of time for the three field variables (across the 1000 test simulations) obtained using KRNO, FNO-3D and LOCA models trained for 200 epochs.

### H.0.5 SHALLOW WATER SIMULATION

We provide some additional numerical results for the shallow water test case. We followed the procedure described in Kovachki et al. (2023) in our numerical studies using FNO. A comparison of results from different models are shown when training is conducted for 200 epochs. The results presented show that FNO-3D performance improves when trained for 200 epochs. We also applied FNO-2D (Kovachki et al., 2023) to learn an autoregressive model that maps the spatio-temporal field at five time instants to the next time step. However, Irrespective of the choice of hyperparameters, we were unable to learn an autoregressive model that provided stable predictions over the testing horizon. Due to this numerical issue, the FNO-2D model is only trained for 100 epochs. Representative predictions from all the models are shown in Figures 14, 15, 16, 17.

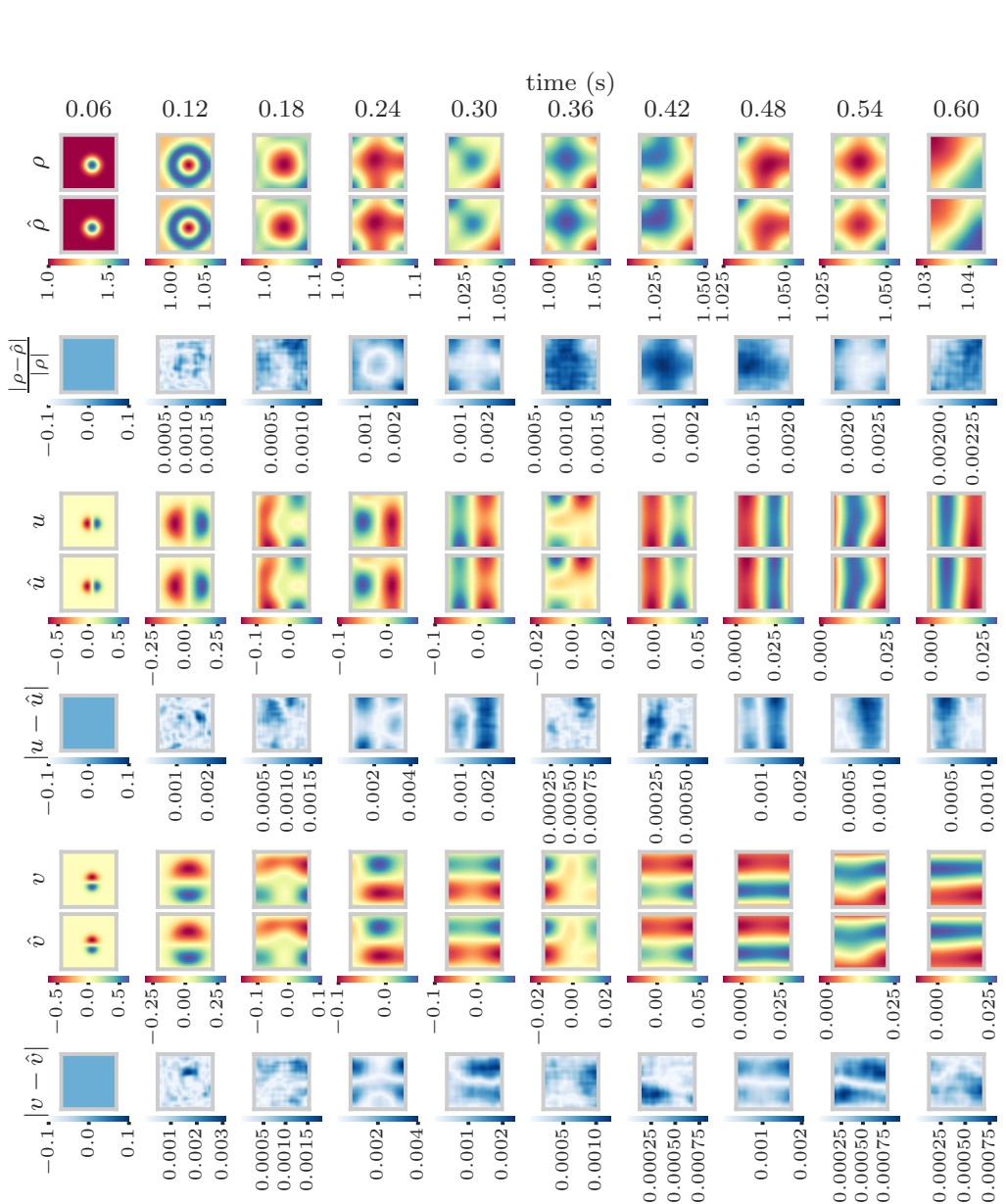

Figure 13: Shallow water problem: Figure shows the predictions $(\hat{\rho}, \hat{u}, \hat{v})$ for the three field variables along with the true fields $(\rho, u, v)$ as a function of time for a test simulation using KRNO trained for 100 epochs.

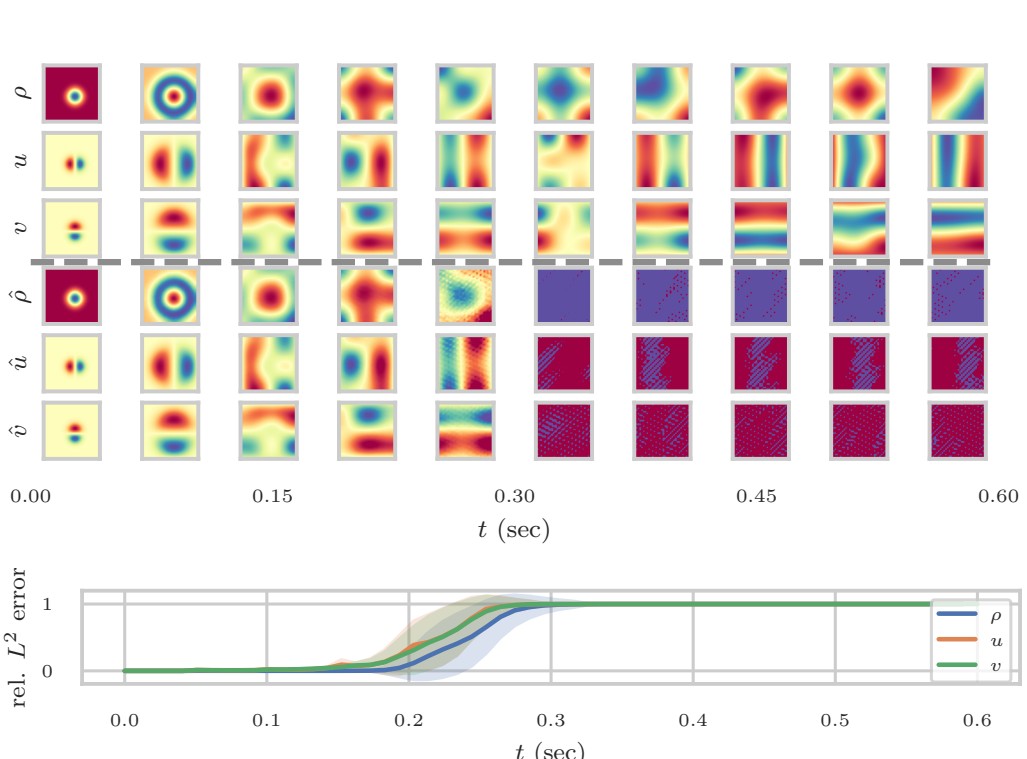

Figure 14: Shallow water problem: Top figure shows the predictions $(\hat{\rho}, \hat{u}, \hat{v})$ for the three field variables along with the true fields $(\rho, u, v)$ as a function of time for a test simulation using FNO-2D trained for 100 epochs. The bottom figure shows the error bars representing the $L^2$ relative errors for three field variables across the 1000 test simulations, with the shaded region indicating $\pm 1$ standard deviation.

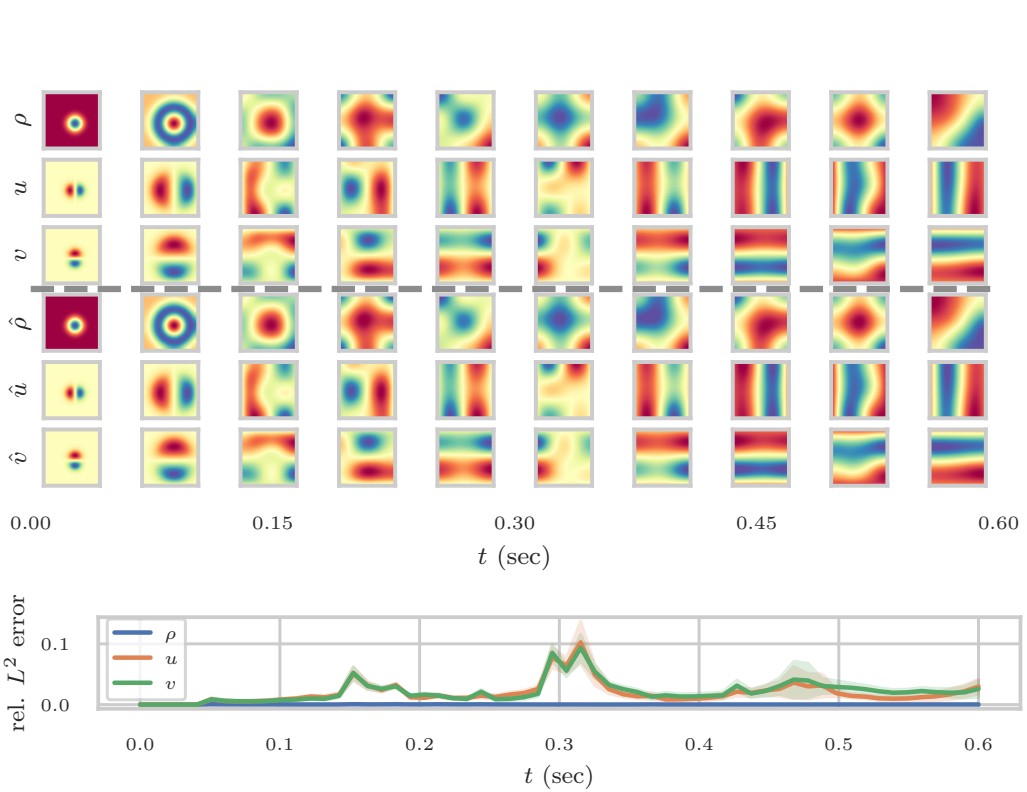

Figure 15: Shallow water problem: Top figure shows the predictions $(\hat{\rho}, \hat{u}, \hat{v})$ for the three field variables along with the true fields $(\rho, u, v)$ as a function of time for a test simulation using KRNO trained for 200 epochs. The bottom figure shows the error bars representing the $L^2$ relative errors for three field variables across the 1000 test simulations, with the shaded region indicating $\pm 1$ standard deviation.

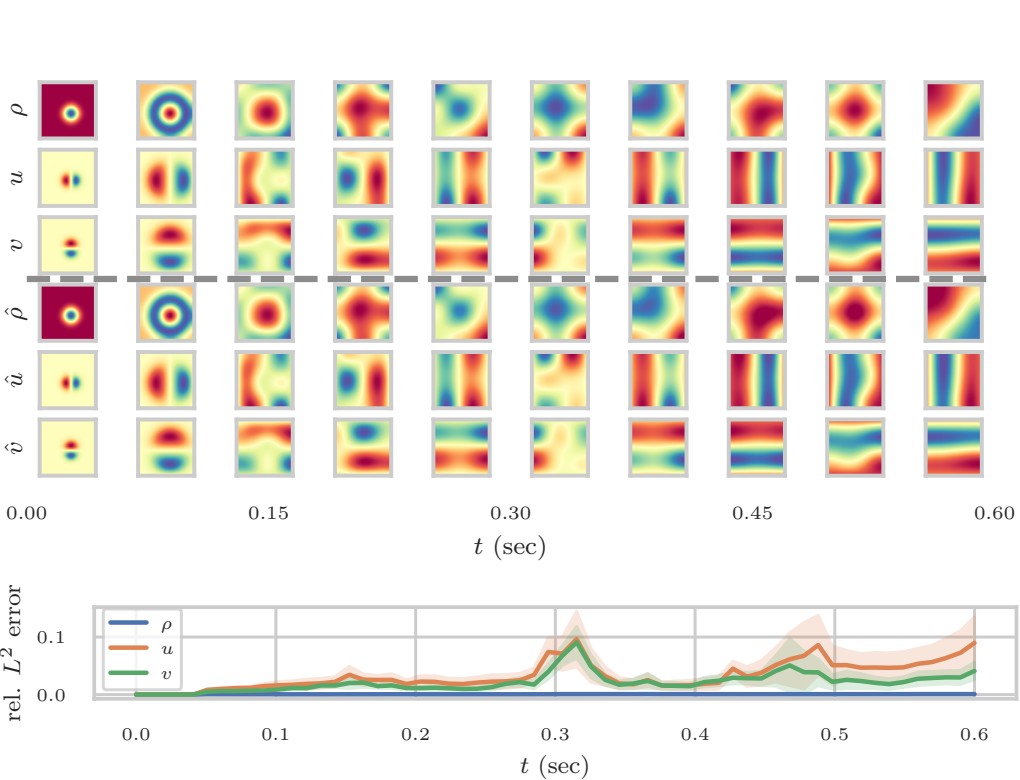

Figure 16: Shallow water problem: Top figure shows the predictions $(\hat{\rho}, \hat{u}, \hat{v})$ for the three field variables along with the true fields $(\rho, u, v)$ as a function of time for a test simulation using FNO-3D trained for 200 epochs. The bottom figure shows the error bars representing the $L^2$ relative errors for three field variables across the 1000 test simulations, with the shaded region indicating $\pm 1$ standard deviation.

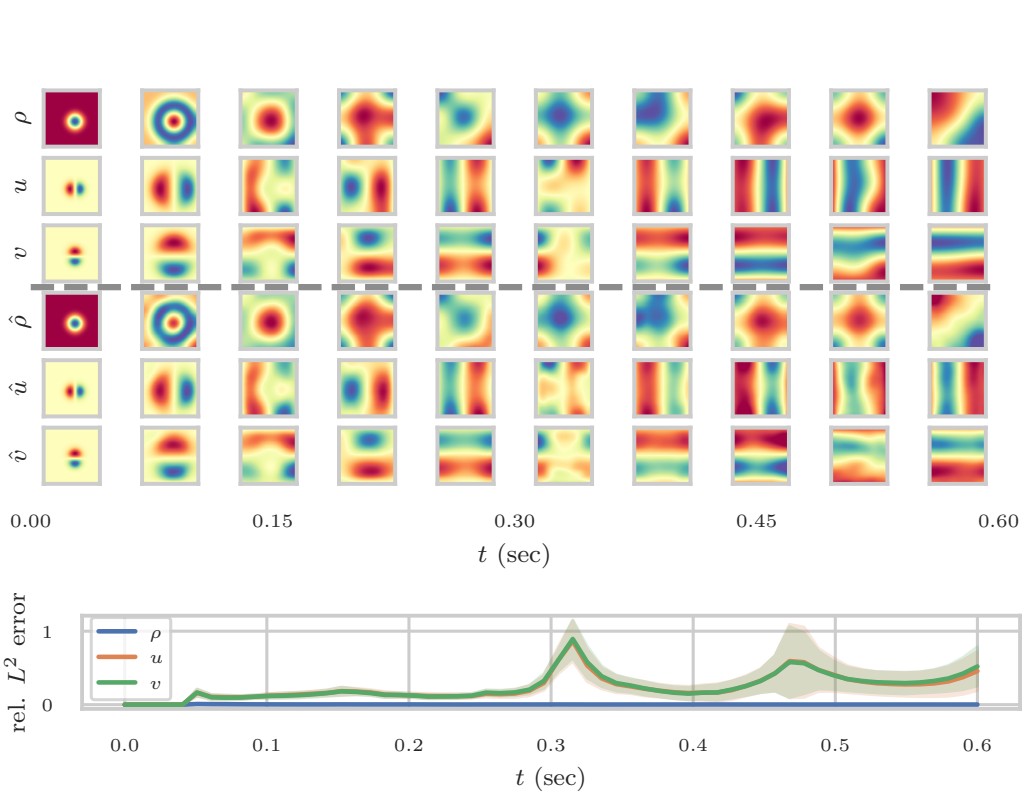

Figure 17: Shallow water problem: Top figure shows the predictions $(\hat{\rho}, \hat{u}, \hat{v})$ for the three field variables along with the true fields $(\rho, u, v)$ as a function of time for a test simulation using LOCA model trained for 200 epochs. The bottom figure shows the error bars representing the $L^2$ relative errors for three field variables across the 1000 test simulations, with the shaded region indicating $\pm 1$ standard deviation.

