# OpenReview forum: "SHIFTING TIME: TIME-SERIES FORECASTING WITH KHATRI-RAO NEURAL OPERATORS"
_ICLR.cc/2025/Conference — ICLR 2025 Conference Withdrawn Submission_

### Official Review · Reviewer_3R22 · 2024-11-01

**Soundness:** 3
**Presentation:** 3
**Contribution:** 2
**Rating:** 5
**Confidence:** 4

**Summary:**

In this manuscript, the authors developed a time series forecasting method based on a neural operator, called Khatri-Raio neural operator (KRNO). In essence, aside from the theoretical development, the proposed KRNO takes a window of time series data as an input and makes a prediction of a future time step. KRNO utilizes a cosine positional embedding to account for the time series data sampled at irregular intervals. KRNO is tested against a set of spatio-temporal problems as well as a few benchmark time series prediction problems. It is shown that KRNO achieves a state-of-the-art performance in many of the benchmark problems.

**Strengths:**

The manuscript is well written. It clearly defines the problem set up, succinctly outline the background, and introduce the method. The proposed method is theoretically robust compared to most of the recent time series papers, which are focused more on engineering aspects.

**Weaknesses:**

One of the major concerns is the applicability of the proposed method for a broader problem. While the proposed neural operator based method is interesting, most of the real data are in essence stochastic processes due to aleatoric uncertainty. For many physics data, the observation noise and natural variability make the observation stochastic. It is unclear if the neural operator theory can be extended to the modeling of such stochastic processes. More over, the model is based on modeling an autonomous system without any exogenous effects, which limits the class of problems that the proposed method can be applied.

One of the major claims of the method is its capability of dealing with irregularly sampled time series data. But in the experiments it is not shown how the proposed method performs with irregularly sampled time series data. The method is based on a numerical approximation of an integral equation. Then, the accuracy of the numerical integration strongly depends on the choice of the quadrature points, which makes the claim about modeling any irregular time series weaker. Depending on the sampling points, it may even induces a numerical instability.

It looks like there are some inconsistencies between the theory and the final model. Probably, it is because some of the terms are not fully explained in the manuscript. See the questions below.

**Questions:**

1. The notation on the right hand-side of Eq (6) is confusing. On LHS, $\kappa((t,X),(t',x'))$ is $R^{q\times p}$. But, on RHS, a similar notation $\kappa((t,X),(t,X))$ is $R^{Nq \times Np}$. Please, check the notations throughout the paper and distinguish between them to avoid a confusion.

2. Up to Eq. (8), the kernel depends only on the quadrature point, i.e., $\kappa((t,x),(t',x'))$. Then, suddenly, in section 2.3, the kernel becomes a function of the input data, $\kappa(U^{t_i}_p)$, which is inconsistent with the theory. If a neural network is used as a nonlinear map between the input data (NOT quadrature point) and output matrix, what's the difference between the proposed neural operator based method and other window based time series models? It becomes very similar to the popular mixer architectures.

3. How do you compute the weights $w$ in Eq (6)? The choice of $w$ is closely related with the accuracy of the numerical integration.

4. The authors claimed that their method can directly learn the "exact kernel". But the "exact kernel" is not a general kernel. It already made a very strong assumption on the kernel structure such that the kernel is based on a set of orthogonal projections of the input space. While the authors made an excuse that such orthogonal decomposition is widely used, for example, in GP, it is used in other previous methods not because it is robust and improves the approximation, but because it makes the problem easier to tackle.

5. How are the error bars in the experiments computed. The proposed method is deterministic. It is not clear what 1,000 test simulation indicates.

---

> ### Author Response · Authors · 2024-11-26
>
> Thanks for your feedback and comments.
>
> > One of the major concerns is the applicability of the proposed method for a broader problem. While the proposed neural operator based method is interesting, most of the real data are in essence stochastic processes due to aleatoric uncertainty. For many physics data, the observation noise and natural variability make the observation stochastic. It is unclear if the neural operator theory can be extended to the modeling of such stochastic processes. More over, the model is based on modeling an autonomous system without any exogenous effects, which limits the class of problems that the proposed method can be applied.
>
> In the present work, we formulate the time-shift operator in the setting of deterministic ordinary/partial differential equations. This formalism enabled sufficient flexibility to provide strong generalization across temporal and spatio-temporal forecasting problems from diverse domains. We agree with your comment that additional extensions to our methodology would enable us to deal with a broader class of problems involving parametrized temporal, spatio-temporal fields, and stochastic processes. In the first case, we would utilize training data containing trajectories for different parameter settings.  It is relatively straightforward to accommodate such additional parameters (accompanying/annotating each trajectory) in the KRNO architecture; for example, we can use a hypernetwork to model the KRNO weights as a function of the parameters. Once the model has been trained, it can be utilized to predict the dynamics for new parameter settings or to estimate the response statistics given a probabilistic model for the parameters.
>
> It is possible in theory to define the time-shift operator for temporal and spatio-temporal stochastic processes. For example, for the case of a temporal stochastic process, one way to proceed would be to define the time-shift operator as $X\_{(t,t_f]} = \mathcal{A}\_{t_p}^{t,t_f} (X_{[t_p,t]}, \xi_{[t_p,t_f]})$, where $X_{[t_p,t]}$ denotes a realization of the process over the time-interval $[t_p,t]$ while $\xi_{[t_p,t_f]}$ denotes the stochastic process driving the dynamics over the input and forecasting time-interval $[t_p,t_f]$. To ensure that the time-shift operator continues to be a causal operator, the integration domain of the kernel integral transform w.r.t $\xi$ should be fixed to $[t_p, \tau]$, where $\tau \in (t, t_f]$ denotes the time instant at which the output function is being predicted.
>
>  Given a training dataset containing the system response for different realizations of the excitation process, the time-shift operator can be learnt using the proposed KRNO architecture.  Further work is needed to study the theoretical properties of the time-shift operator of temporal and spatio-temporal stochastic processes (governed by a stochastic differential equation or a stochastic partial differential equation) under appropriate regularity assumptions.
>
> It is worth noting that the notion of time-shift operator proposed in our work can be generalized to forced dynamical systems in a continuous time or continuous space-time setting similar to the approach outlined previously for stochastic processes. For example, for the case of ODEs,
> the forced time-shift operator can be defined as $X\_{(t,t_f]} = \mathcal{A}\_{t_p}^{t,t_f} (X_{[t_p,t]}, f_{[t_p,t_f]})$, where $X_{[t_p,t]}$ denotes the state trajectory over the time-interval $[t_p,t]$ while
> $f_{[t_p,t_f]}$ denotes the forcing function over the input and forecasting time-interval $[t_p,t_f]$. To ensure that the time-shift operator is causal (i.e., the predictions made at time instant $\tau$ are not influenced by the future values of the forcing function), the integration domain of the kernel integral transform w.r.t $f$ should be set to $[t_p, \tau]$, where $\tau \in (t, t_f]$. A similar approach can be used to define the time-shift operator for forced spatio-temporal dynamical systems. These extensions would enable our approach to systems with exogenous effects.
>
>
> > One of the major claims of the method is its capability of dealing with irregularly sampled time series data. But in the experiments it is not shown how the proposed method performs with irregularly sampled time series data. The method is based on a numerical approximation of ....
>
> We now include a numerical study demonstrating the performance of our architecture on datasets with highly irregular(random) sampling patterns; please see Appendix-F of the revised paper. We note that our current KRNO implementation based on the mid-point/trapezoidal quadrature rule performs reasonably well when we use random sampling to generate the training trajectory. As expected, the predictive accuracy can deteriorate if the samples within the time-window adjacent to the testing interval is highly sparse. In Appendix-F, we discuss this along with how predictive accuracy can be improved using alternative quadrature schemes.

---

> ### Author Response · Authors · 2024-11-26
>
> >**Q1.** The notation on the right hand-side of Eq (6) is confusing. On LHS,
> $ \kappa (\\{t,X\\},\\{t', x'\\})$ is $\mathbb{R}^{q \times p}$. But, on RHS, a similar notation $ \kappa (\\{t,X\\},\\{t,X\\})$ is $\mathbb{R}^{Nq \times Np}$. Please, check the notations throughout the paper and distinguish between them to avoid a confusion.
>
> We apologize for the inconsistency in our notation and we are grateful that you brought this to our attention. To avoid the confusion with $t$, we now use $\bar{t} = \\{t_1, \ldots, t_n\\}$ to refer to the set of quadrature points in the time domain and $X$ to denote the set of points in the full spatio-temporal Cartesian product grid ${{\bar{t}} \times x^{(1)} \times \dots x^{(d-1)}}$; please see Equation (6) and line 187 onwards in the revised paper.
>
> In Equation (6), $\kappa (X,\\{t', x'\\}) \in \mathbb{R}^{Nq \times p}$ represents the kernel evaluated between all the quadrature nodes $X$ in the output domain and a single node $\\{t', x'\\}$ in the input domain. Meanwhile, $ \kappa (X, X) \in \mathbb{R}^{Nq \times Np}$ is the kernel evaluated between all the quadrature nodes $X$ in the output and input domains.
>
> Please note that for simplicity of exposition, the main text discusses the KRNO integral transform layer for the special case when the input and output functions are defined over the same domain and the quadrature nodes are shared. In the revised paper, we provide additional details with clearer notation to show how the integral transform layers can be generalized to account for situations when the input and output functions are defined over different domains (with each domain using a different set of quadrature nodes). Please see Appendix C in the revised paper which shows how the KRNO integral transform layers can be generalized to  map a vector field defined over $\mathcal{U}(\Omega\_\ell \times \mathcal{I}\_\ell;R^p)$ to $\mathcal{U}(\Omega\_{\ell+1} \times \mathcal{I}_{\ell+1};R^q)$. The generalized KRNO integral transform layer can be viewed as a continuous analog of upsampling and downsampling layers used in convolutional neural networks. We hope that this generalized description improves clarity, and we look forward for your feedback and suggestions.
>
>
> >**Q2.** Up to Eq. (8), the kernel depends only on the quadrature point, i.e., $ \kappa (\\{t,X\\},\\{t', x'\\})$. Then, suddenly, in section 2.3, the kernel becomes a function of the input data, $\kappa(U_p^{t_i})$ , which is inconsistent with the theory. If a neural network is used as a nonlinear map between the input data (NOT quadrature point) and output matrix, what's the difference between the proposed neural operator based method and other window based time series models? It becomes very similar to the popular mixer architectures.
>
> Thank you for raising this point. Section 2.3 outlines some practical aspects of training KRNO models on temporal and spatio-temporal problems. Please note that as mentioned in the theory section, the arguments to the kernel continue to be either time or space-time coordinates. The kernels are consistently evaluated at quadrature nodes across input and output domains at each layer, as outlined in Section 2.2.  To avoid potential confusion, we updated the notation used for the parametrized time-shift operator appearing in the loss functions defined in Section 2.3.
>
> Unlike discrete-time autoregressive window-based models, which assume fixed sampling frequencies for the observations, our approach adopt a continuous space-time approach. This generalized formulation enables flexible training and forecasting at arbitrary resolutions, setting it apart from typical mixer architectures that lack this flexibility when dealing with spatial-temporal datasets.
>
> >**Q3.** How do you compute the weights $w$ in Eq (6)? The choice of is closely related with the accuracy of the numerical integration.
>
> In our current implementation, the weights $w$ in Equation (6) were computed using midpoint quadrature for the temporal dimension and trapezoidal quadrature for the spatial dimensions.  For the test cases we considered, we found that our current implementation was able to learn a time-shift operator capable of capturing complex spatio-temporal interactions. In the revised paper, we discuss this point in more detail and discuss alternative quadrature schemes that can improve efficiency and memory requirements for high-resolution datasets and improve predictive accuracy when the training data sampling is highly irregular; please see Appendix F and G of the revised paper.

---

> ### Author Response · Authors · 2024-11-26
>
> >**Q4.** The authors claimed that their method can directly learn the "exact kernel". But the "exact kernel" is not a general kernel. It already made a very strong assumption on the kernel structure such that the kernel is based on a set of orthogonal projections of the input space. While the authors made an excuse that such orthogonal decomposition is widely used, for example, in GP, it is used in other previous methods not because it is robust and improves the approximation, but because it makes the problem easier to tackle.
>
> Thank you for briging up this important point.  We agree that  product structured kernels are widely used in GP modelling since it can provide computational advantages and not because it leads to better generalization.
>
> In neural operator architectures, we are using a composition of multiple kernel integral transform layers stacked with point-wise lifting and projection layers. Thanks to the universal approximation theorem for neural operators [1], this enables us to approximate arbitrary maps between two separable Banach spaces. To reemphasize, the product structured kernel doesn't impose any theoretical limitations in the context of operator learning. A particularly attractive feature of KRNO is that it allows us to learn a flexible *non-stationary* kernel for each component of the product structured kernel. As evidenced by our numerical studies, this additional flexibility provides performance gains over stationary kernel-based neural operator architectures and also significantly reduces the number of parameters.
>
> We acknowledge that the term 'exact kernel' used in our original submission is confusing in the operator learning setting. What we originally meant to convey is that when numerically estimating convolution integrals, the only source of error in our methodology comes from the quadrature scheme; in other words, the non-stationary kernel is evaluated "exactly" without introducing any additional approximation errors. We clarify this point in the revised manuscript; please see page-3, line-161.
>
>
> >**Q5.** How are the error bars in the experiments computed. The proposed method is deterministic. It is not clear what 1,000 test simulation indicates.
>
> Thank you for bringing this up. While our method is deterministic, the shallow water problem includes 1,000 distinct test simulations, each tracking the time evolution of fluid column height $\rho$, velocities $v_1$, and $v_2$
> for different initial conditions. The error bars represent the variation in model error across these 1,000 simulations, computed as a function of time, to indicate the robustness of the model's performance across varied scenarios.
>
> [1] Kovachki, Nikola, et al. "Neural operator: Learning maps between function spaces with applications to pdes." Journal of Machine Learning Research 24.89 (2023): 1-97.

---

### Official Review · Reviewer_SUFw · 2024-11-02

**Soundness:** 4
**Presentation:** 3
**Contribution:** 3
**Rating:** 6
**Confidence:** 3

**Summary:**

This paper addresses the challenges of irregular sampling problems in autoregressive models. A Khatri-Rao neural operator is proposed that 1) allows continuous relaxation of the discrete lag factor for irregular sampling and 2) defines non-stationary integral transforms for better model flexibility. Empirical results show improved performance on a range of benchmark problems compared to baseline models.

**Strengths:**

1. The problem of irregular sampling in time-series forecasting is important.
2. The idea of the continuous time-shift operator is creative and the methodology is well-structured.
3. Experiments were performed on temporal and spatio-temporal problems to present the effectiveness of the proposed method.

**Weaknesses:**

1. The authors mentioned the time-shift operator maps the entire continuous history of the past time-window into its future values. Could the authors discuss the benefit of this neural operator versus neural controlled differential equations [1] and its variants?
2. The presentation in Section 3 needs to be better. In Figure 2 and Figure 4, the authors could provide a relative error similar to Figure 5 to better present the prediction accuracy. Figure 2 and Figure 4 are not referred to in the main text. In Figure 4, the authors should mark which variables are ground truths and which are predictions. Also, could the authors explain what causes the large fluctuations of certain variables in Figure 4 and Figure 5 error bars? Table 5 seems to show that the time resolution affects the performance of the proposed method. Could the authors further explain that?
3. The ablation study is not sufficient. The authors should consider comparing the time complexity of the proposed method with Graph Neural operator, Multipole Graph Neural Operator, and Fourier Neural Operator. Also, the effect of time resolution and spatial complexity should also be addressed.

[1] Kidger, Patrick, et al. "Neural controlled differential equations for irregular time series." Advances in Neural Information Processing Systems 33 (2020): 6696-6707.

**Questions:**

Please find the questions in the weaknesses section above.

---

> ### Author Response · Authors · 2024-11-26
>
> Thanks for your feedback and comments.
>
> > The authors mentioned the time-shift operator maps the entire continuous history of the past time-window into its future values. Could the authors discuss the benefit of this neural operator versus neural controlled differential equations [1] and its variants?
>
> Thank you for bringing up this point. In order to apply neural ODE approaches to spatio-temporal modeling problems, spatial discretization is necessary. Furthermore, in practice, an autoencoder is needed (i.e., a latent-ODE modeling perspective) to ensure that the dimensionality of state-space learned by the neural ODE is reasonable. The performance of neural ODEs when applied to spatio-temporal problems will therefore be dependent on the spatial discretization scheme and the autoencoder architecture. In contrast to neural ODEs, neural operator approaches can be directly applied to spatio-temporal problems without  requiring a spatial discretization step.
>
> It is worth noting that the notion of time-shift operator proposed in our work can be generalized to forced dynamical systems in a continuous time or continuous space-time setting. For example, for the case of ODEs,
> the forced time-shift operator can be defined as $ X_{(t,t_f]} = \mathcal{A}\_{t_p}^{t,t_f} (X_{[t_p,t]}, f_{[t_p,t_f]}) $, where $X_{[t_p,t]}$ denotes the state trajectory over the time-interval $[t_p,t]$ while
> $f_{[t_p,t_f]}$ denotes the forcing function over the input and forecasting time-interval $[t_p,t_f]$. To ensure that the time-shift operator is causal (i.e., the predictions made at time instant $\tau$ are not influenced by the future values of the forcing function), the integration domain of the kernel integral transform w.r.t $f$ should be set to $[t_p, \tau]$, where $\tau \in (t, t_f]$. This generalization can be leveraged to learn the time-shift operator of a latent forced dynamical system (similar to neural CDEs). For spatio-temporal modeling applications, we could similarly model a latent spatio-temporal process that captures the dynamics of the observed process. It is not clear to us at this point if a latent representation would provide better generalization compared to our current approach that directly models the time-shift operator of the observed process. However, this perspective has the potential to provide gains in efficiency during training and inference for high-resolution observations.
>
> > The presentation in Section 3 needs to be better. In Figure 2 and Figure 4, the authors could provide a relative error similar to Figure 5 to better present the prediction accuracy. Figure 2 and Figure 4 are not referred to in the main text. In Figure 4, the authors should mark which variables are ground truths and which are predictions. Also, could the authors explain what causes the large fluctuations of certain variables in Figure 4 and Figure 5 error bars? Table 5 seems to show that the time resolution affects the performance of the proposed method. Could the authors further explain that?
>
> We thank the reviewer for this constructive feedback. We have made the following revisions to address these concerns.
>
> - We have incorporated relative error plots in Figures 2 and 4 and included the updated figures in the Appendix; please see Figures 9 and 12 of the revised paper. The caption for Figure 4 has been updated to indicate which variables represent ground truths and which are predictions for better clarity.
>
> - Regarding the fluctuations in error bars, in Figure 12, the relative error for the shallow water problem exhibits higher fluctuations in velocity fields $u$ and $v$ where the true values are close to zero. In such cases, even small absolute differences between the predicted and true values result in significant relative errors. For Figure 5, we believe that the fluctuations in the relative error for temperature $T$ are due to difficulties in capturing some of the complex seasonal patterns on a global scale.
>
> - Regarding the performance on the M4 dataset, we believe that KRNO is able to generalize better on datasets such as Weekly and Daily since these datasets have no clear trends and seasonality [2].
>
>
> [1] Kidger, Patrick, et al. "Neural controlled differential equations for irregular time series." Advances in Neural Information Processing Systems 33 (2020): 6696-6707.
>
> [2] Wang, Rui, et al. "Koopman neural operator forecaster for time-series with temporal distributional shifts." The Eleventh International Conference on Learning Representations. 2023.

---

> > ### Author Response · Authors · 2024-11-26
> >
> > > The ablation study is not sufficient. The authors should consider comparing the time complexity of the proposed method with Graph Neural operator, Multipole Graph Neural Operator, and Fourier Neural Operator. Also, the effect of time resolution and spatial complexity should also be addressed.
> >
> > Thank you for your suggestion to include additional numerical studies illustrating how the time complexity of different approaches compare. In the revised paper, we provide numerical studies comparing the training and inference complexity of KRNO and FNO-3D across various spatial resolutions for the shallow water dataset; please see Appendix-G. To summarize, the parameter count of KRNO is independent of the size of the quadrature grid while the parameter count of FNO-3D depends on the number of Fourier modes. For the shallow water test case, the parameter count of KRNO is 145,319 while the parameter count of FNO-3D is 5.5 million for 12 modes. Note that when the number of modes is set to 64, the parameter count of FNO-3D increases to 157 million, the memory usage increases to 10,324 MB and training time per iteration (0.1725 seconds) exceeds KRNO.
> >  In Appendix F, we also discuss how the memory footprint and time complexity of KRNO can be further reduced.

---

> > > ### Comment · Reviewer_SUFw · 2024-11-27
> > >
> > > Thanks for the authors' effort to improve the presentation of the work. All my concerns have been addressed. I raise my scores to 6.

---

### Official Review · Reviewer_R62i · 2024-11-04

**Soundness:** 3
**Presentation:** 3
**Contribution:** 2
**Rating:** 3
**Confidence:** 4

**Summary:**

This work introduces a continuous time-shift operator for time-series forecasting. Using Khatri-Rao neural operators, they claim to achieve efficient handling of irregular sampling and high-resolution forecasting in spatio-temporal settings, with results that are claimed to compete strongly with current methods.

**Strengths:**

It is an interesting take on the topic and I appreciate tackling the spatio-temporal forecasting problems. I like the choice of datasets they are rather interesting. Appeal to Koopman operator theory is quite welcome.

**Weaknesses:**

A lot of the background theory behind the methodology seems like there is some intentional obfuscation of the reader going on. This can be seen in the proof in Appendix A which is a very standard application of Gronwall's Lemma which can be found in most ODE textbooks (the authors use William F Ames and BG Pachpatte. Inequalities for differential and integral equations, volume 197), so really one should just cite its use rather than present a one page proof for it. To be clear I'm not against (re)showing of very standard proofs, but there should be signifanctly good reason, such as branching off of an intermediate step and so on. It is somewhat like presenting a "proof for the law of large numbers" when one should really in principle cite it as is common practice. Because this occurs it leads me to suspect a lot of the mathematical formulation shown, and overall notation  in this paper is just there for an intentional obfuscation for the reader. What could be intersting in this case, for example, would be if one could charactise the nature of constant C, otherwise there is not much point.

As a result this (what is suspected to be) intentional mathematical obfuscation, Lines 85 - 107 can more or less be written in 3-4 lines with the same properties as follows:
"Consider an ODE z˙(t) = F(z(t)), z(0) = z0 with Lipschitz continuous F: Rn → Rn over t ∈ [0,T]. While the classical flow map solution maps only initial conditions to solutions, we define a time-shift operator At,tf tp that maps the history of z over [tp,t] to its future values over (t,tf], where 0 ≤ tp < t < tf ≤ T. At,tf tp is continuous and satisfies the semigroup property: At2,tf tp = At2,tf t1 ◦ At1,t2 tp . Unlike discrete autoregressive models, this continuous-time formulation naturally handles irregularly sampled data without requiring adjoint sensitivity calculations as in neural ODEs."

Similarly I feel there is no need for Proposition 1 or its associated proof. The authors do state: "Proposition 1 follows from similar results for Kronecker structured GP regression (Saatçi, 2011)" -- which gets the main point across essentially since it is not "similar" but rather more or less the same since by definition the "Khatri-Rao product *is* the Kroneckor product just generalized to block matrix structures. See the first 1-2 lines of: https://en.wikipedia.org/wiki/Khatri%E2%80%93Rao_product
I simply cannot see the benefit of dedicating then, a lengthy proposition, and a proof in App B, unless I may be missing something.

As a result of establishing this link the pytorch implementation of the Khatri-Rao product doesn't seem completely necessary (Appendix C), since this is a standard implementation in many python / pytorch libraries on tensor algebra which are very, very optimized already, so I don't fully get its explicit novelty of why one would present the code for it as a novelty (or at least why there is a whole section dedicated to it - unless I am missing something). See:
https://tensorly.org/dev/modules/generated/tensorly.tenalg.khatri_rao.html

Given that Appendix A - C do not seem to be genuine novelties from what I can tell, Appendix D appears to be the crux of the novelty  but it is only a few lines long, and rather than push for any meaningful novelty it gets relegated to a reference unfortunately.

Moreover Figure 8 (Appendix D), the forecasts for GasRateCo2, Wooly, HeartRate seem a little bit worrying in selling the applicability of this model to real world data.

All in all, I think rather than finished work I think this could be the potential of a good start. May I suggest submitting these initial results to a workshop where the ideas can be foundationally built upon and further developed perhaps.

**Questions:**

Since the questions section is meant for persuading the reviewer (of which I am open), I may defer to the addressing of points in the above "weakness" section, of which quite a few points have been raised.

---

> ### Author Response · Authors · 2024-11-26
>
> Thanks for your feedback and comments.
>
> > A lot of the background theory behind the methodology seems like there is some intentional obfuscation of the reader going on. This can be seen in the proof in Appendix A which is a very standard application of Gronwall's Lemma which can be found in most ODE textbooks (the authors use William F Ames and BG Pachpatte. Inequalities for differential and integral equations, volume 197), so really one should just cite its use rather than present a one page proof for it. To be clear I'm not against (re)showing of very standard proofs, but there should be signifanctly good reason, such as branching off of an intermediate step and so on. It is somewhat like presenting a "proof for the law of large numbers" when one should really in principle cite it as is common practice. Because this occurs it leads me to suspect a lot of the mathematical formulation shown, and overall notation in this paper is just there for an intentional obfuscation for the reader. What could be intersting in this case, for example, would be if one could charactise the nature of constant C, otherwise there is not much point.
>
> To the best of our knowledge, our paper is the first study into learning the continuous time-shift operator from time-series and spatio-temporal data. Therefore, we believe that it is important from a theoretical viewpoint to characterize the key properties of this new operator. The continuity proof turns out to be straightforward (we use Grönwall's lemma which we are fully aware is a text-book  result that is ubiquitous in any paper on analysis of numerical schemes for ordinary/partial differential equations). Even though the proof is fairly straightforward, we included it in the Appendix A of our original submission for completeness.
>
> We have now revised our continuity lemma/proof  to make it more general and as compact as possible. In addition, we now also quantify the continuity constant as suggested; please see page 15 of Appendix A in the revised paper. In the updated proof, we now show that the continuity property holds under standard Lipschitz assumptions without the need for the implicit regularity assumption in the lemma presented in our original submission.
> We have kept the continuity proof in the Appendix as before and mention it in a sentence in the main text; please see Page 2, line 94 of the revised paper.
>
> We hope that we have addressed your concerns and we look forward to any additional feedback you may have.
>
>
> > As a result this (what is suspected to be) intentional mathematical obfuscation, Lines 85 - 107 can more or less be written in 3-4 lines with the same properties as follows: "Consider an ODE z˙(t) = F(z(t)), z(0) = z0 with Lipschitz continuous F: Rn → Rn over t ∈ [0,T]. While the classical flow map solution maps only initial conditions to solutions, we define a time-shift operator At,tf tp that maps the history of z over [tp,t] to its future values over (t,tf], where 0 ≤ tp < t < tf ≤ T. At,tf tp is continuous and satisfies the semigroup property: At2,tf tp = At2,tf t1 ◦ At1,t2 tp . Unlike discrete autoregressive models, this continuous-time formulation naturally handles irregularly sampled data without requiring adjoint sensitivity calculations as in neural ODEs."
>
>
> Thank you for your suggestions on improving our writeup and making it more compact. As suggested, we have significantly shortened the preamble to our methodology in the revised paper; please see Section 2.1 of the revised paper.

---

> ### Author Response · Authors · 2024-11-26
>
> > Similarly I feel there is no need for Proposition 1 or its associated proof. The authors do state: "Proposition 1 follows from similar results for Kronecker structured GP regression (Saatçi, 2011)" -- which gets the main point across essentially since it is not "similar" but rather more or less the same since by definition the "Khatri-Rao product is the Kroneckor product just generalized to block matrix structures. See the first 1-2 lines of: https://en.wikipedia.org/wiki/Khatri%E2%80%93Rao_product I simply cannot see the benefit of dedicating then, a lengthy proposition, and a proof in App B, unless I may be missing something.
>
> As mentioned after introducing equation (6), the computational complexity associated with approximating kernel integral transform layers scales as $\mathcal{O}(N^2)$, where $N$ is the number of quadrature nodes.
> Proposition 1 shows how the computational complexity can be reduced to $\mathcal{O}(N^{1 + 1/d})$. To the best of our knowledge, this is a **not a known result** although the proof is fairly straightforward. Given the practical impact of this result for application areas beyond spatio-temporal forecasting, we state it as a proposition in the main text with the proof provided in the Appendix. Similar to the continuity proof, we cite sources for the standard results that we draw upon in our proof.
>
> We have now generalized our theoretical result to space-time kernel integral transform layers that map a vector field defined over $ \mathcal{U}(\Omega\_\ell \times \mathcal{I}\_\ell; R^p) $ to $ \mathcal{U}(\Omega\_{\ell+1} \times \mathcal{I}\_{\ell+1}; R^q) $;
> please see Appendix C in the revised manuscript. This generalization enables us to tackle a broader class of operator learning problems where the inputs and outputs are defined over different spatio-temporal domains (with each domain using a different set of quadrature nodes) while maintaining the original $\mathcal{O}(N^{1 + 1/d})$ complexity. The generalized KRNO integral transform layer can be viewed as a continuous analog of upsampling and downsampling layers used in convolutional neural networks. In addition, as discussed in Appendix C of the revised paper, this also provides the flexibility to design memory-efficient neural operators by varying the resolution of the functions in the hidden layers.
>
> > As a result of establishing this link the pytorch implementation of the Khatri-Rao product doesn't seem completely necessary (Appendix C), since this is a standard implementation in many python / pytorch libraries on tensor algebra which are very, very optimized already, so I don't fully get its explicit novelty of why one would present the code for it as a novelty (or at least why there is a whole section dedicated to it - unless I am missing something). See: https://tensorly.org/dev/modules/generated/tensorly.tenalg.khatri_rao.html
>
> Efficient implementation of our proposed KRNO architecture requires computation of general Khatri-Rao products, with support for batched operations; please see Section 2.2 for more details.
>
> The Khatri-Rao product implementation in TensorLy is limited to the **special case** involving column-wise Kronecker products of 2D tensors (matrices) and lacks support for batched operations which is crucial for scaling KRNO to large-scale spatio-temporal datasets. This is also the case for other open-source implementations for Khatri-Rao products. This is not surprising since Khatri-Rao products are not that widely encountered in machine learning architectures -- one exception we are aware of is scalable Gaussian process modeling; see, for example, [1,2].
>
> We, therefore, put together a new implementation that can efficiently compute general Khatri-Rao products while providing the flexibility for handling batch processing and variable input-output resolutions. We included a code snippet in the Appendix since we believe that our implementation can be leveraged in diverse applications such as compressive sensing and generative modelling of spatio-temporal signals. We acknowledge that additional optimizations to our implementation may enable improvements in memory access patterns and efficiency on GPUs -- we leave this for future work.
>
> [1] Nickson, Thomas, et al. "Blitzkriging: Kronecker-structured stochastic Gaussian processes." arXiv preprint arXiv:1510.07965 (2015).
>
> [2] Evans, T., and Nair, P.B. "Scalable Gaussian processes with grid-structured eigenfunctions (GP-GRIEF)." ICML 2018.

---

> ### Author Response · Authors · 2024-11-26
>
> > Given that Appendix A - C do not seem to be genuine novelties from what I can tell, Appendix D appears to be the crux of the novelty but it is only a few lines long, and rather than push for any meaningful novelty it gets relegated to a reference unfortunately.
>
> We believe that the main novelty of our work lies in the continuous time-shift operator that we learn in an operator-inference setting for temporal and spatio-temporal modeling, and a new architecture for operator learning (KRNO) that leverages highly flexible non-stationary kernels with a computational complexity of $\mathcal{O}(N^{1 + 1/d})$. To the best of our knowledge, the proposed operator learning architecture is the first approach which achieves almost linear cost while enabling non-stationary integral transforms. We believe that this presents a significant advance in neural operator learning with applications to a diverse range of problems.
>
> Appendices A-C contains additional details supporting the main ideas/advances presented in the main body of the text.
> Please note that Appendix-D (Appendix-E in the revised paper) is brief since here we provide additional details on how the components of the product structured KRNO kernel introduced in equation (7) can be parameterized in practice.
>
>
> > Moreover Figure 8 (Appendix D), the forecasts for GasRateCo2, Wooly, HeartRate seem a little bit worrying in selling the applicability of this model to real world data.
>
>
> We would like to point out that, out of the eight datasets in the Darts collection, GasRateCo2, Wooly, and HeartRate datasets pose significant challenges due to the pronounced distribution shift between training and testing data. This is the reason we opted to include prediction plots for these datasets in the Appendix. It is worth noting that other state-of-the-art methods also struggle with these datasets. We would like to bring to your attention that we have evaluated the performance of our method using 27 test cases and compared against 22 modern approaches for temporal and spatio-temporal forecasting problems; see Section 3. It would be highly unusual for a single method to do well on all the test cases. Nonetheless, our method demonstrated strong generalizability, achieving top performance on 8/27 test cases and ranking within the top 3 for 19/27 test cases.

---

### Official Review · Reviewer_Q6rb · 2024-11-04

**Soundness:** 3
**Presentation:** 3
**Contribution:** 3
**Rating:** 6
**Confidence:** 3

**Summary:**

This paper presents a novel approach for time series forecasting by learning a continuous time-shift operator with the Khatri-Rao Neural Operator (KRNO). Unlike traditional autoregressive models that require regular sampling intervals, KRNO handles irregularly sampled data and can forecast in high resolution across both time and space. Through extensive testing on temporal and spatio-temporal datasets, the authors demonstrate that KRNO performs competitively against state-of-the-art methods in terms of scalability, accuracy, and computational efficiency.

**Strengths:**

1. Robustness to Irregular Sampling: KRNO effectively addresses the challenge of irregularly sampled data, making it applicable in real-world scenarios where regular sampling isn’t guaranteed.
2. Computational Efficiency: The model achieves almost linear computational costs while allowing for exact, non-stationary kernel evaluations, which is highly advantageous for large-scale spatio-temporal problems.
3. High-Resolution Forecasting: KRNO’s ability to generate super-resolution forecasts in both time and space allows it to capture fine-grained details that are valuable in fields like climate modeling and medical monitoring.
4. Comprehensive Benchmarking: The paper includes thorough experiments across various benchmark datasets, highlighting KRNO’s performance in comparison to existing models.

**Weaknesses:**

- Complexity and Resource Demand: KRNO’s architecture and training process are complex and may require substantial computational resources, which could limit its accessibility and practical deployment.
- Generalization Across Data Types: While KRNO performed well on the tested datasets, it’s unclear how well it generalizes to data with distinct irregular patterns or high variability, which may affect its robustness.
- Hyperparameter Sensitivity: The need for extensive hyperparameter tuning (such as for tp and tf) in KRNO might increase the complexity of implementation, especially in cases where computational resources are limited.

**Questions:**

- How does KRNO handle irregular patterns or noise in datasets that differ significantly from those used in the experiments?
- While the authors claim near-linear computational costs, could more specific benchmarks on processing time and resource usage for high-resolution spatio-temporal data be provided?
- Are there specific industry applications where KRNO’s advantages would be most beneficial, or is it generally applicable across various domains?

---

> ### Author Response · Authors · 2024-11-26
>
> Thanks for your feedback and comments.
>
> >**Q1.** How does KRNO handle irregular patterns or noise in datasets that differ significantly from those used in the experiments?
>
> To evaluate KRNO's ability to handle irregular patterns and noisy data, we tested it on the challenging M4 dataset, which includes 100,000 univariate time series from various domains such as finance and demographics. The dataset features varying levels of irregularity and noise across six frequency-based subsets (hourly to yearly), reflecting diverse real-world scenarios. KRNO performed competitively, ranking among the top two methods in subsets like weekly and daily series with minimal seasonality. These results, detailed in Section 3.2 and Table 5, highlight KRNO's capacity to generalize effectively to noisy, irregular datasets.
>
>
> >**Q2.** While the authors claim near-linear computational costs, could more specific benchmarks on processing time and resource usage for high-resolution spatio-temporal data be provided?
>
> Thank you for your suggestion to include additional numerical studies illustrating how the time complexity of different approaches compare. In the revised paper, we provide numerical studies comparing the training and inference complexity of KRNO and FNO-3D across various spatial resolutions for the shallow water dataset; please see Appendix-G. To summarize, the parameter count of KRNO is independent of the size of the quadrature grid while the parameter count of FNO-3D depends on the number of Fourier modes. For the shallow water test case, the parameter count of KRNO is 145,319 while the parameter count of FNO-3D is 5.5 million for 12 modes. Note that when the number of modes is set to 64, the parameter count of FNO-3D increases to 157 million, the memory usage increases to 10,324 MB and training time per iteration (0.1725 seconds) exceeds KRNO.
>  In Appendix F, we also discuss how the memory footprint and time complexity of KRNO can be further reduced.
>
>
> >**Q3.** Are there specific industry applications where KRNO’s advantages would be most beneficial, or is it generally applicable across various domains?
>
> While the proposed methodology is versatile and applicable across various domains, our numerical studies indicate that KRNO excels particularly in spatial and spatio-temporal problems. For instance, its ability to handle irregular sampling and forecast at super-resolution makes it highly suitable for applications such as climate modeling, fluid dynamics simulations, and geospatial analysis. These domains are expected to benefit significantly from KRNO's capacity to efficiently model complex spatio-temporal dynamics.
> Additionally, KRNO’s inherent capability to map between varying resolutions using kernel integral transform layers provides a natural foundation for problems involving high-resolution spatio-temporal datasets, further expanding its utility in industry applications. It is also worth noting that the KRNO architecture can be used for classification and regression problems where the training inputs are images/signals with varying resolution/sampling-frequency. We leave this for future work.
>
> > Generalization Across Data Types: While KRNO performed well on the tested datasets, it’s unclear how well it generalizes to data with distinct irregular patterns or high variability, which may affect its robustness.
>
>
> We now include a numerical study demonstrating the performance of our architecture on datasets with highly irregular(random) sampling patterns; please see Appendix-F of the revised paper. We note that our current KRNO implementation based on the mid-point/trapezoidal quadrature rule performs reasonably well when we use random sampling to generate the training trajectory. As expected, the predictive accuracy can deteriorate if the samples within the time-window adjacent to the testing interval is highly sparse. In Appendix-F, we discuss this along with how predictive accuracy can be improved using alternative quadrature schemes.
>
>
> > Hyperparameter Sensitivity: The need for extensive hyperparameter tuning (such as for tp and tf) in KRNO might increase the complexity of implementation, especially in cases where computational resources are limited.
>
> KRNO can be viewed as a continuous relaxation of the discrete lag factor used in standard autoregressive models. The optimal choice of hyperparameters $t_p$ and $t_f$ depends on the length scales of the dynamics of the system being studied. In our experience, finding a reasonable setting for these hyperparameters is fairly straight forward since they are continuous variables. We would also like to mention here that further work is needed to explore the possibility of training a single model on a dataset containing input/output trajectories for different settings of $t_p$ and $t_f$. This would enable the possibility of learning a model that can predict the dynamics at different lengthscales.

---

> > ### Comment · Reviewer_Q6rb · 2024-11-28
> >
> > Thanks for the answers, and all of my concerns have been addressed. I will maintain my score 6.

---

### Author Response · Authors · 2024-11-26

We sincerely thank all reviewers for their thorough evaluations and constructive feedback. We have carefully addressed each comment and revised our paper to incorporate the suggested improvements. Below, we summarize the key changes and responses to concerns:

 - We have edited Sections 1 and 2 to make it more concise, improve clarity, and fix notational issues.

 - In Appendix A of the revised paper, we provide a more general proof for the continuity of the time-shift operator that holds without any additional regularity assumptions beyond Lipschitz continuity. In addition, we also specify the expression for the continuity constant in terms of the problem parameters.

 - We have now generalized our theoretical result to space-time kernel integral transform layers that map a vector field defined over $\mathcal{U}(\Omega\_\ell \times \mathcal{I}\_\ell;R^p)$ to $\mathcal{U}(\Omega\_{\ell+1} \times \mathcal{I}\_{\ell+1};R^q)$; please see Appendix C in the revised paper for more details. This generalization enables us to tackle a broader class of operator learning problems where the inputs and outputs are defined over different spatio-temporal domains (with each domain using a different set of quadrature nodes) while maintaining the original $\mathcal{O}(N^{1 + 1/d})$ complexity.  The generalized KRNO integral transform layer can be viewed as a continuous analog of upsampling and downsampling layers used in convolutional neural networks.

  - We now include a numerical study demonstrating the performance of our architecture on datasets with irregular sampling patterns; please see Appendix F for more details.

 - In the revised paper, we include a new numerical study comparing the training and inference costs of KRNO and FNO-3D across various spatial resolutions; please see Appendix G for more details.

---

### Note · Authors · 2025-01-26

I have read and agree with the venue's withdrawal policy on behalf of myself and my co-authors.